# Lipid polarity gradient formed by ω-hydroxy lipids in tear film prevents dry eye disease

Masatoshi Miyamoto, Takayuki Sassa, Megumi Sawai, Akio Kihara*

Laboratory of Biochemistry, Faculty of Pharmaceutical Sciences, Hokkaido University, Sapporo, Japan

**Abstract** Meibum lipids form a lipid layer on the outermost side of the tear film and function to prevent water evaporation and reduce surface tension. (O-Acyl)-ω-hydroxy fatty acids (OAHFAs), a subclass of these lipids, are thought to be involved in connecting the lipid and aqueous layers in tears, although their actual function and synthesis pathway have to date remained unclear. Here, we reveal that the fatty acid ω-hydroxylase *Cyp4f39* is involved in OAHFA production. *Cyp4f39*-deficient mice exhibited damaged corneal epithelium and shortening of tear film break-up time, both indicative of dry eye disease. In addition, tears accumulated on the lower eyelid side, indicating increased tear surface tension. In *Cyp4f39*-deficient mice, the production of wax diesters (type 1ω and 2ω) and cholesteryl OAHFAs was also impaired. These OAHFA derivatives show intermediate polarity among meibum lipids, suggesting that OAHFAs and their derivatives contribute to lipid polarity gradient formation for tear film stabilization.

## Introduction

The tear film maintains visual function by eliminating foreign materials, supplying oxygen and nutrients to the ocular surface (cornea and conjunctiva), and reducing friction between the eyelid and the ocular globe (*Ohashi et al., 2006*). Tear film consists of three layers: in order from the outside, these are the tear film lipid layer (TFLL), the aqueous layer, and the glycocalyx layer (*Figure 1A*; *Gipson, 2004*; *Cwiklik, 2016*). The TFLL contributes to the suppression of water evaporation from the aqueous layer and reduces the surface tension of tears (*Butovich, 2013*). Since the TFLL lipids are secreted from the meibomian glands, which are distributed behind the eyelids, they are collectively called meibum lipids. The aqueous layer contains nutrients, electrolytes, and bioactive molecules. The mucin-rich glycocalyx layer has a role in maintaining the aqueous layer on the corneal surface.

Dry eye disease is caused by destabilization of the tear film and is accompanied by symptoms of eye discomfort and visual dysfunction, and potentially by ocular surface damage (*Gayton, 2009*). The prevalence of dry eye disease varies among countries and regions (7–33% of the population) and is increasing year by year (*Gayton, 2009*; *Dana et al., 2019*). Dry eye disease is roughly classified into two types: aqueous-deficient dry eye (ADDE) and evaporative dry eye (EDE) (*Bron and Tiffany, 2004*; *Craig et al., 2017*). Most cases of EDE are caused by meibomian gland dysfunction (MGD). MGD is the most common cause of dry eye disease: one study reported that 87% of dry eye patients suffer from MGD (either MGD alone or MGD with ADDE) (*Horwath-Winter, 2003*).

The TFLL is thought to consist of two sublayers, a nonpolar lipid sublayer and an amphiphilic lipid sublayer (*Butovich, 2011*; *Green-Church et al., 2011*; *Butovich, 2017*). The two most abundant meibum lipids are cholesteryl esters (CEs) and wax esters (WEs) (*Figure 1A*), the total amount of which varies among reports but is 60–96% of total meibum lipids (*Lam et al., 2011*; *Chen et al.,*

*For correspondence:
kihara@pharm.hokudai.ac.jp

**Competing interests:** The authors declare that no competing interests exist.

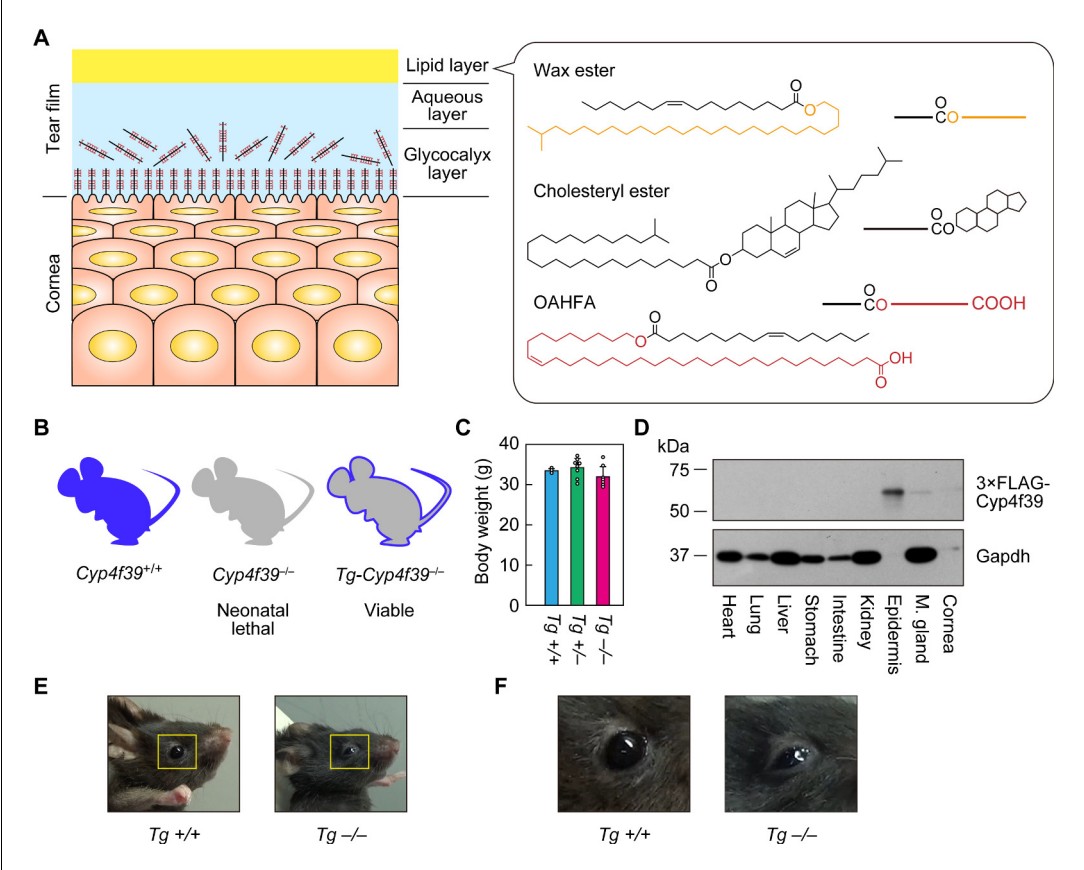

**Figure 1.** Generation of *Tg-Cyp4f39⁻/⁻* mice. (A) Schematic illustration of the cornea, tear film, and structures of major meibum lipids (wax ester [WE], cholesteryl ester [CE], and (*O*-Acyl)-ω-hydroxy fatty acid [OAHFA]), together with their simplified structures. Orange and red indicate the fatty alcohol (FAl) and ω-OH fatty acid (FA) moieties, respectively. (B) Schematic diagram of *Cyp4f39⁺/⁺* (wild type), *Cyp4f39⁻/⁻* (*Cyp4f39* whole-body knockout), and *Tg-Cyp4f39⁻/⁻* mice. Blue represents *Cyp4f39* expression and gray indicates *Cyp4f39* deficiency. (C) Body weight of *Tg-Cyp4f39⁺/⁺* (n = 3), *Tg-Cyp4f39⁺/⁻* (n = 8), and *Tg-Cyp4f39⁻/⁻* (n = 7) mice at 10–12 months of age. Values presented are means ± SD. (D) Total lysates (20 μg) prepared from the heart, lung, liver, stomach, small intestine, kidney, epidermis, meibomian gland, and cornea of 12-month-old *Tg-Cyp4f39⁻/⁻* mice were separated by SDS-PAGE and subjected to immunoblotting with anti-FLAG and anti-GAPDH antibodies (loading control). (E, F) Photographs of 12-month-old *Tg-Cyp4f39⁺/⁺* and *Tg-Cyp4f39⁻/⁻* mice. The photographs in (F) are enlargements of the areas enclosed by the yellow frames in panel (E). Tg +/+, *Tg-Cyp4f39⁺/⁺*; Tg–/–, *Tg-Cyp4f39⁻/⁻*; M. gland, Meibomian gland.

*2013*; *Butovich, 2017*). These lipids are highly nonpolar and form the nonpolar lipid sublayer. The major components of the amphiphilic lipid sublayer are (*O*-acyl)-ω-hydroxy fatty acids (OAHFAs) (1– 5%) and cholesteryl OAHFAs (Chl-OAHFAs; ~3%), which are the cholesterol adducts of OAHFAs (*Butovich, 2017*). The amphiphilic lipid sublayer also contains triglycerides (~1%), fatty acids (FAs; 0.1–1%), phospholipids (<0.1%), and cholesterol (<0.5%) (*Butovich, 2017*). It is postulated that this sublayer functions in stabilizing tear film by producing an interface between the nonpolar lipid layer, which constitutes most of the TFLL, and the aqueous layer beneath it (*Butovich et al., 2009*). OAH-FAs have a structure in which a C30–C36 ω-hydroxy (ω-OH) FA and a C16–C18 FA are ester-linked (*Figure 1A*; *Butovich et al., 2009*; *Butovich, 2017*). However, the biosynthesis pathway of OAHFAs and the genes involved in their synthesis are still unknown. Furthermore, a mouse model that is unable to produce OAHFAs has not yet been created. Therefore, the actual function of OAH-FAs in tear film has not yet been elucidated.

We previously revealed that the cytochrome P450 members CYP4F22 (human) and its mouse ortholog Cyp4f39 exhibit high ω-hydroxylase activity toward ≥C30 FAs (*Ohno et al., 2015*; *Miyamoto et al., 2020*). *CYP4F22* mutations cause the skin disease congenital ichthyosis, and *Cyf4f39* knockout (*Cyp4f39⁻/⁻*) in mice is neonatal lethal because of impaired skin permeability barrier formation (*Lefèvre et al., 2006*; *Miyamoto et al., 2020*). The epidermis has a special class of

ceramides, ω-O-acylceramides, which is essential for skin permeability barrier function (*Kihara, 2016*), and *CYP4F22*/*Cyp4f39* mutations cause failure to produce ω-O-acylceramides (*Ohno et al., 2015*; *Miyamoto et al., 2020*). ω-O-Acylceramides contain ≥C30 ω-OH FAs, as seen in OAHFAs. Therefore, we hypothesize that CYP4F22/Cyp4f39 are also involved in FA ω-hydroxylation in the OAHFA biosynthesis pathway. Whole-body *Cyf4f39* gene knockout is neonatal lethal (*Miyamoto et al., 2020*), so we created *Cyp4f39$^{-/-}$ Tg* (*IVL-Cyp4f39*) mice (hereafter, *Tg-Cyp4f39$^{-/-}$*), in which the epidermal barrier defect was rescued by the transgenic expression of *Cyp4f39* in the epidermis. Using these mice, we examined the involvement of Cyp4f39 in the production of OAHFAs and OAHFA derivatives, and their roles in tear film stabilization.

## Results

### Palpebral ptosis and abnormal tear covering on the eyeball surface in *Cyp4f39*-deficient mice

To examine the involvement of *Cyp4f39* in OAHFA production in the meibomian glands, the neonatal lethality caused by whole-body *Cyp4f39* disruption must be circumvented. For this purpose, we created *Tg-Cyp4f39$^{-/-}$* mice, in which all tissues except the epidermis lacked *Cyp4f39* expression, by expressing a *3×FLAG* tagged *Cyp4f39* transgene under the control of the epidermis-specific involucrin (*IVL*) promoter (*Figure 1B*). *Tg-Cyp4f39$^{-/-}$* mice grew normally to adulthood, and their body weights were comparable to those of *Tg-Cyp4f39$^{+/+}$* and *Tg-Cyp4f39$^{+/-}$* mice (*Figure 1C*). To confirm the epidermis-specific expression of 3×FLAG-Cyp4f39 protein, total cell lysates were prepared from several tissues of *Tg-Cyp4f39$^{-/-}$* mice and subjected to immunoblotting with anti-FLAG antibody. 3×FLAG-Cyp4f39 was highly expressed in the epidermis, and weak expression was observed in the cornea and meibomian glands (*Figure 1D*). It is known that the *IVL* promoter is active in the cornea (*Adhikary et al., 2005*). The detection of 3×FLAG-Cyp4f39 protein in the meibomian gland is probably due to expression in the keratinized epithelial cells that constitute the meibomian gland ducts rather than to the production of meibum lipids by meibocytes. Palpebral ptosis was observed in *Tg-Cyp4f39$^{-/-}$* mice (*Figure 1E,F*). In addition, tears did not spread normally over the entire eyeball surface in these mice, but rather accumulated on the lower eyelid side, suggesting that tear film surface tension was higher than normal.

### *Cyp4f39* deficiency causes dry eye with plugging of the meibomian gland orifices

We have previously reported that mice lacking the FA elongase gene *Elovl1* show dry eye accompanied by palpebral ptosis and increased eye blinking, due to shortening of the chain length of meibum lipids (*Sassa et al., 2018*). An increase in blinking frequency has also been reported among human dry eye patients (*Su et al., 2018*). Blinking frequency was measured in 1–12-month-old *Tg-Cyp4f39$^{-/-}$* mice, and we found that it was greatly increased relative to blinking frequency in *Tg-Cyp4f39$^{+/+}$* and *Tg-Cyp4f39$^{+/-}$* mice: the average frequencies in *Tg-Cyp4f39$^{+/+}$* and *Tg-Cyp4f39$^{+/-}$* mice were <0.6 times/min at all ages, whereas in *Tg-Cyp4f39$^{-/-}$* mice it was 3.3, 7.4, 7.3, and 11.3 times/min at 1–3, 4–6, 7–9, and 10–12 months, respectively (*Figure 2A*). There were no significant differences in these frequencies among the different ages, suggesting that dry eye does not progress with age.

In dry eye disease patients, tear film break-up time (BUT) is shortened due to tear film destabilization (*Tsubota, 2018*). To measure BUT in *Tg-Cyp4f39$^{-/-}$* mice, fluorescein solution was loaded onto the eye surface and observed under a slit lamp microscope. The average BUT in the *Tg-Cyp4f39$^{+/+}$* mice was 7.4 s, whereas that in *Tg-Cyp4f39$^{-/-}$* mice was 2.5 s, ~1/3 of that in *Tg-Cyp4f39$^{+/+}$* mice (*Figure 2B*). We then scored corneal epithelial damage and found that *Tg-Cyp4f39$^{-/-}$* mice scores were ~2.6 times higher than those of *Tg-Cyp4f39$^{+/+}$* mice (*Figure 2C*). These results indicate that *Tg-Cyp4f39$^{-/-}$* mice exhibit a dry eye phenotype with tear film destabilization and corneal epithelial damage.

Dry eye disease is classified as ADDE or EDE, depending on the cause of the pathology (*Bron and Tiffany, 2004*). The majority of EDE is caused by MGD and is often accompanied by obstruction at the orifices of the meibomian glands. Obstruction of meibomian gland orifices with white, semi-liquid plugging was observed in all *Tg-Cyp4f39$^{-/-}$* mice (8 out of 8 mice) examined

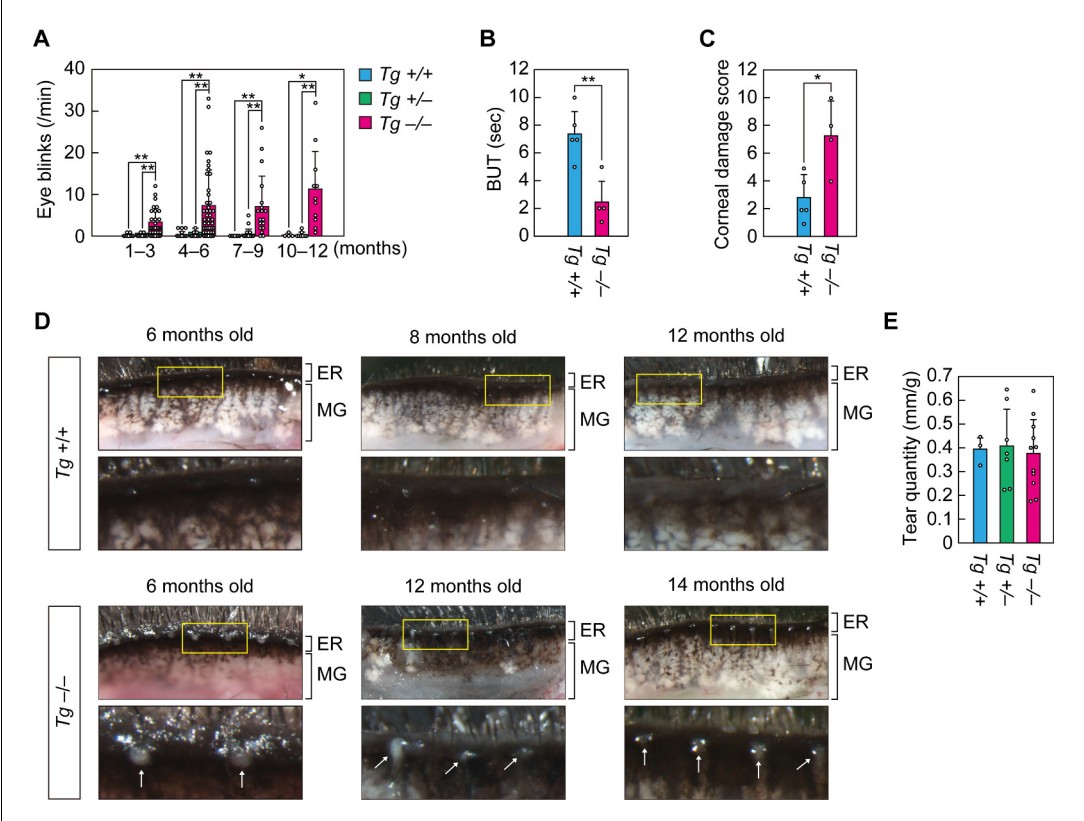

**Figure 2.** *Cyp4f39*-deficient mice exhibit dry eye. (**A**) Blink frequency was measured in 1–12-month-old *Tg-Cyp4f39*$^{+/+}$ (n = 7), *Tg-Cyp4f39*$^{+/-}$ (n = 9), and *Tg-Cyp4f39*$^{-/-}$ (n = 14) mice. The total number of measurements for mice at each age were: *Tg-Cyp4f39*$^{+/+}$ mice, 11 at 1–3 months, 19 at 4–6 months, 10 at 7–9 months, 4 at 10–12 months; *Tg-Cyp4f39*$^{+/-}$ mice, 12 at 1–3 months, 24 at 4–6 months, 18 at 7–9 months, 8 at 10–12 months; and *Tg-Cyp4f39*$^{-/-}$ mice, 35 at 1–3 months, 45 at 4–6 months, 18 at 7–9 months, and 11 at 10–12 months. Values presented are means of blink rate per min ± SD (*, p<0.05; **, p<0.01; Tukey-Kramer's test). Break-up time (BUT) (**B**) and corneal damage score (**C**) were measured for 8–17-month-old *Tg-Cyp4f39*$^{+/+}$ (n = 3) and *Tg-Cyp4f39*$^{-/-}$ (n = 4) mice. Experiments were performed on both eyes, and 5 and 4 measurements were obtained from *Tg-Cyp4f39*$^{+/+}$ and *Tg-Cyp4f39*$^{-/-}$mice, respectively. Values presented are means ± SD (*, p<0.05; **, p<0.01; Student's *t*-test). (**D**) Upper eyelids from 6–12-month-old *Tg-Cyp4f39*$^{+/+}$ mice and 6–14-month-old *Tg-Cyp4f39*$^{-/-}$ mice and photographed under a light microscope. The lower images are enlarged views of the areas surrounded by yellow rectangles in the upper images. The white arrows represent obstruction of the meibomian gland orifice. (**E**) Tear quantity was measured in 10–12-month-old *Tg-Cyp4f39*$^{+/+}$ (n = 3), *Tg-Cyp4f39*$^{+/-}$ (n = 7), and *Tg-Cyp4f39*$^{-/-}$ (n = 11) mice using the phenol red-thread test. Values presented are means ± SD (*, p<0.05; Student's *t*-test). Tg +/+, *Tg-Cyp4f39*$^{+/+}$; Tg +/–, *Tg-Cyp4f39*$^{+/-}$; Tg –/–, *Tg-Cyp4f39*$^{-/-}$; ER, eyelid rim; MG, meibomian gland.

between the ages of 6–17 months (6 months, two mice; 12 months, three mice; 14 months, one mouse; and 17 months, two mice) (*Figure 2D*). The degree of plugging did not appear to change with age. On the other hand, no such plugging was observed in any *Tg-Cyp4f39*$^{+/+}$ mice examined between the ages of 6–12 months (0 out of 9 mice). The meibomian glands of *Tg-Cyp4f39*$^{-/-}$ mice were not swollen compared to those of the control mice. There were no differences in tear volume among *Tg-Cyp4f39*$^{+/+}$, *Tg-Cyp4f39*$^{+/-}$, and *Tg-Cyp4f39*$^{-/-}$ mice (*Figure 2E*). On the basis of these results, we conclude that *Tg-Cyp4f39*$^{-/-}$ mice exhibit EDE with meibomian gland plugging.

## Normal formation of meibomian glands and cornea and normal expression of meibum lipid-related genes in *Cyp4f39*-deficient mice

To examine the effects of *Cyp4f39* deficiency on the formation of meibomian glands and the differentiation of meibocytes, histological analyses were conducted on the meibomian glands of *Tg-Cyp4f39*$^{-/-}$ mice by hematoxylin/eosin staining. The meibomian gland is composed of acini that are connected to ducts. Meibocytes propagate in the basal parts of the acini, then differentiate and move toward the center of the acini while synthesizing and accumulating meibum lipids. Finally, the

matured meibocytes collapse and become enucleated dead cells, releasing the accumulated meibum lipids (holocrine secretion mode) (*Knop et al., 2011*). In the acini of both *Tg-Cyp4f39*[+/+] and *Tg-Cyp4f39*[−/−] mice, regularly arranged meibocytes in the basal parts and enucleated cells in the central parts were observed (*Figure 3A*), indicating that meibocyte differentiation and maturation are normal in *Tg-Cyp4f39*[−/−] mice. We then performed histological analyses on the cornea. This structure is composed of three layers: epithelium, stroma, and endothelium. The two groups of mice showed almost no difference in cornea thickness, number of layers, or morphology (*Figure 3B*). Thus, cornea formation is also normal in *Tg-Cyp4f39*[−/−] mice.

We next examined the expression levels of meibum lipid-related genes (*Awat1*, *Awat2*, *Far1*, *Far2*, *Soat1*, and *Cyp4f39*) in *Tg-Cyp4f39*[−/−] mice by real-time quantitative RT-PCR. The acyl-CoA wax alcohol acyltransferases Awat1 and Awat2 synthesize WEs from an acyl-CoA and a fatty alcohol (FAl) (*Turkish et al., 2005*). The fatty acyl-CoA reductases Far1 and Far2 catalyze the production of FAls (*Cheng and Russell, 2004*). The sterol *O*-acyltransferase Soat1 is involved in CE production in the meibomian glands (*Meiner et al., 1996*; *Yagyu et al., 2000*). There were no differences in the expression levels of any of these genes between *Tg-Cyp4f39*[+/+] and *Tg-Cyp4f39*[−/−] mice (*Figure 3C*). These results indicate that *Cyp4f39* deficiency does not affect the expression of meibum lipid-related genes other than *Cyp4f39*.

## *Cyp4f39* deficiency causes a decrease in C16:1 OAHFA levels in meibum lipids

To investigate whether *Cyp4f39* is involved in the production of OAHFAs in the meibomian glands, we first tried to establish a specific detection method for OAHFAs using mass spectrometry (MS). Since an OAHFA standard was not commercially available, we chemically synthesized OAHFA [(*O*-C18:1)-ω-OH C22:0 FA] from ω-OH behenic acid (C22:0 FA) and oleoyl (C18:1) chloride (*Figure 4—figure supplement 1*). The synthesized OAHFA was subjected to liquid chromatography (LC) coupled with tandem MS (MS/MS) analysis using the product ion scanning mode. We detected two fragment ions: one with a mass-to-charge ratio (*m/z*) value of 281.06, corresponding to [C18:1 FA−H]⁻, and one with an *m/z* value of 354.95, corresponding to [M−H−(C18:1 FA−OH)]⁻ (*Figure 4A,B*), confirming proper synthesis.

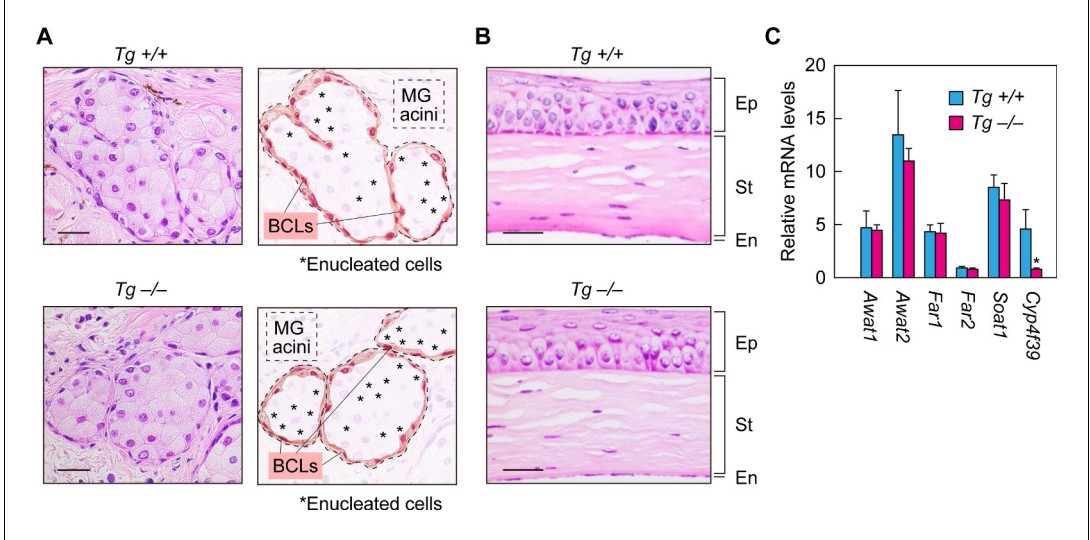

**Figure 3.** Normal formation of meibomian glands and cornea and normal gene expression in *Cyp4f39*-deficient mice. (**A, B**) Paraffin sections of 12-month-old *Tg-Cyp4f39*[+/+] and *Tg-Cyp4f39*[−/−] mice were stained with hematoxylin and eosin. Bright-field images of meibomian glands (A, left panels) and cornea (B) photographed under a light microscope, and schematic diagrams of the meibomian gland acini (A, right panels) are presented. Scale bar, 25 µm. (**C**) Total RNAs were prepared from the meibomian glands of 12-month-old *Tg-Cyp4f39*[+/+] (n = 3) and *Tg-Cyp4f39*[−/−] (n = 3) mice and subjected to real-time quantitative RT-PCR using specific primers for *Awat1*, *Awat2*, *Far1*, *Far2*, *Soat1*, *Cyp4f39* , or the housekeeping gene *Hprt*. Values are amounts of each mRNA relative to that of *Hprt* and represent means ± SD (*, p<0.05; Student's *t*-test). *Tg +/+, Tg-Cyp4f39*[+/+]; *Tg −/−, Tg-Cyp4f39*[−/−]; MG, meibomian gland; BCL, basal cell layer; Ep, epithelium; St, stroma; En, endothelium.

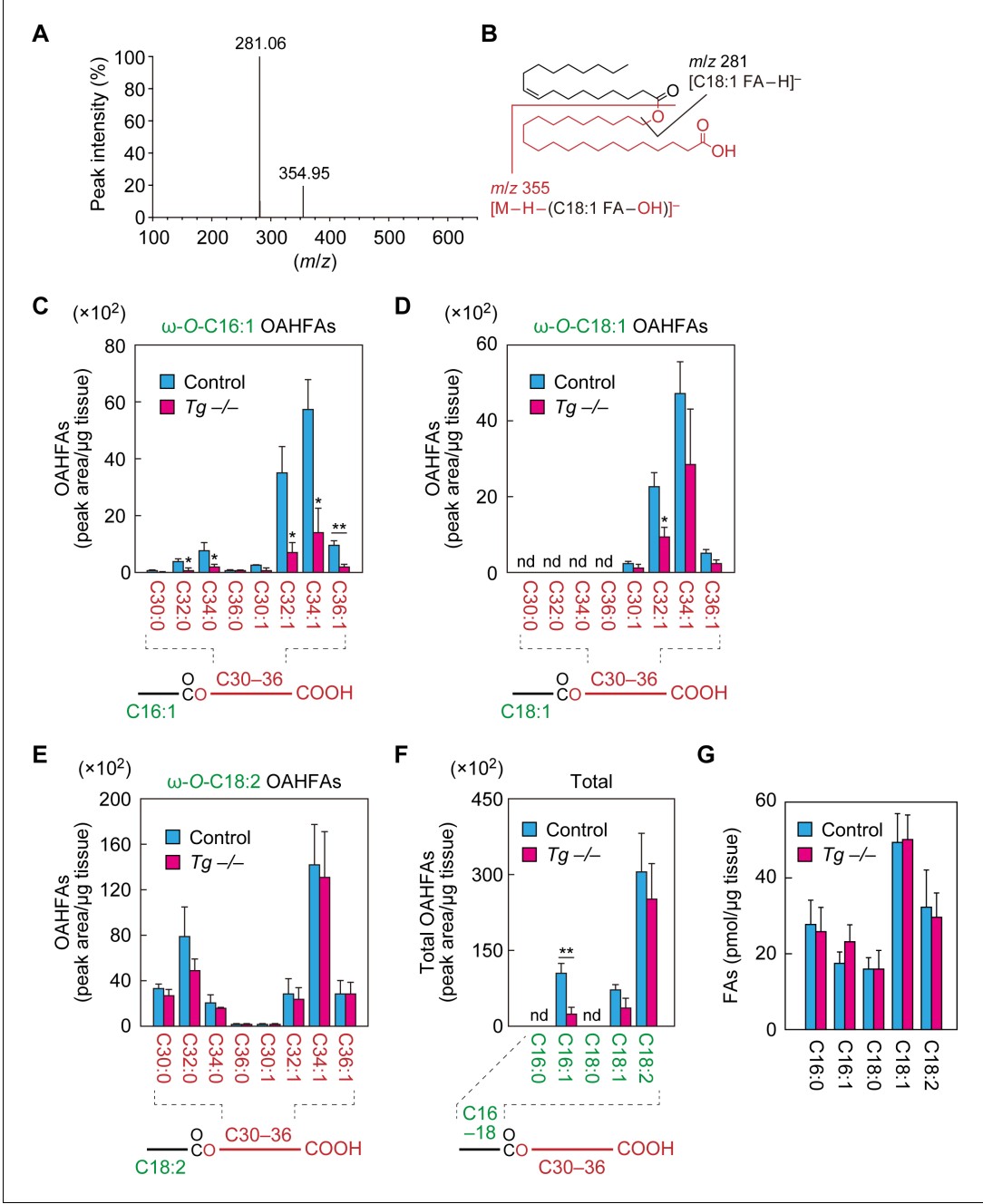

**Figure 4.** Reduction of C16:1 OAHFAs in meibomian glands from *Cyp4f39*-deficient mice. (**A, B**) Product ion scanning of the OAHFA (*O*-C18:1)-ω-OH C22:0 FA was performed by LC-MS/MS by selecting the [M–H]⁻ ion, with *m/z* 619.5, as a precursor ion. The MS spectrum (**A**) and the predicted cleavage positions (**B**) are shown. A synthesis scheme for OAHFA [(*O*-C18:1)-ω-OH C22:0 FA] is provided in *Figure 4—figure supplement 1*. (**C–F**) Lipids were extracted from the meibomian glands of 12-month-old control (*Tg-Cyp4f39*⁺/⁺ [n = 2] and *Tg-Cyp4f39*⁺/⁻ [n = 1]) and *Tg-Cyp4f39*⁻/⁻ (n = 3) mice. After derivatization with AMPP, OAHFAs containing C16:0 (**F**), C16:1 (**C, F**), C18:0 (**F**), C18:1 (**D, F**), and C18:2 (**E, F**) FA were analyzed by LC-MS/MS. The peak areas for OAHFA species that have different FA chain lengths and degrees of unsaturation (saturated or monounsaturated; panels [**C–E**]), and their total amounts (**F**) are shown. Values presented are means ± SD (*, p<0.05; **, p<0.01; Student's *t*-test). The simplified structure for each OAHFA is shown below the graph. (**G**) Lipids extracted from the meibomian glands of 6–12-month-old *Tg-Cyp4f39*⁺/⁺ (n = 3) and *Tg-Cyp4f39*⁻/⁻ mice (n = 3) were subjected to alkaline treatment and AMPP derivatization, and C16:0, C16:1, C18:0, C18:1, and C18:2 FAs were quantified using LC-MS/MS. Values are presented as means ± SD. nd, not detected; *Tg* –/–, *Tg-Cyp4f39*⁻/⁻.

*Figure 4 continued on next page*

*Figure 4 continued*

The online version of this article includes the following figure supplement(s) for figure 4:

**Figure supplement 1.** Synthesis scheme of OAHFA ([*O*-C18:1]-ω-OH C22:0 FA).

Next, we developed a highly sensitive method for detection of OAHFAs, in which OAHFAs were derivatized to *N*-(4-aminomethylphenyl)pyridinium (AMPP) and analyzed by LC-MS/MS in multiple reaction monitoring (MRM) mode. We measured OAHFA levels in the meibomian glands of control and *Tg-Cyp4f39*$^{-/-}$ mice. In control mice, OAHFAs containing a C16:1, C18:1, or C18:2 FA (C16:1 OAHFA, C18:1 OAHFA, or C18:2 OAHFA) and an ω-OH C30–36 FA were detected, but none containing C16:0 or C18:0 FA were detected (*Figure 4C–F*). Among the C16:1 OAHFAs, more monounsaturated ω-OH FAs were detected than saturated ones, and ω-OH C34:1 FA was the most abundant (*Figure 4C*). Among the C18:1 OAHFAs, only monounsaturated ω-OH FAs were detected, with ω-OH C34:1 FA again the most abundant (*Figure 4D*). Among the C18:2 OAHFAs, both saturated and monounsaturated ω-OH FAs were detected, with ω-OH C32:0 FA and ω-OH C34:1 FA the most abundant, respectively (*Figure 4E*).

In *Tg-Cyp4f39*$^{-/-}$ mice, the amounts of C16:1 OAHFAs were lower than in control mice, regardless of chain length and degree of unsaturation, and the total amount was ~20% of that of control mice (*Figure 4C,F*). Regarding C18:1 OAHFAs, only the fraction containing ω-OH C32:1 FA was significantly lower in *Tg-Cyp4f39*$^{-/-}$ mice than in the control (*Figure 4D*). Although the total amount of C18:1 OAHFAs was somewhat lower in *Tg-Cyp4f39*$^{-/-}$ mice, this difference was not significant (*Figure 4F*). Neither the amount of any C18:2 OAHFA species nor the total amount of OAHFAs differed between control and *Tg-Cyp4f39*$^{-/-}$ mice (*Figure 4E,F*). Thus, *Cyp4f39* deficiency had no or almost no effect on C18:2 and C18:1 OAHFA levels. However, as these OAHFAs exist in the epidermis (*Hirabayashi et al., 2019*), they may have been derived from the keratinized epithelial cells that constitute the meibomian gland ducts or from epidermis cells that contaminated the samples, rather than from meibocytes producing meibum lipids. Therefore, we speculate that C16:1 OAHFAs, which were reduced in *Tg-Cyp4f39*$^{-/-}$ mice, are the major meibum lipids in mice.

In the OAHFAs in the meibomian glands of control mice, C16:1 FA was more abundant than C18:1 FA (*Figure 4F*). To determine whether the abundance of FAs present in the meibomian glands was reflective of the OAHFA composition, we next measured the amounts of FAs in the meibomian glands of *Tg-Cyp4f39*$^{+/+}$ mice. The levels of C18:1 FA were highest, followed by those of C18:2, C16:0, C16:1, and C18:0 FA in descending order (*Figure 4G*). This result suggests that the higher levels of C16:1 OAHFA compared to those of C18:1 OAHFA were due to the substrate specificity of the unknown acyltransferase that produces OAHFAs in mice. The FA levels in *Tg-Cyp4f39*$^{-/-}$ mice were not significantly different from those in *Tg-Cyp4f39*$^{+/+}$ mice.

## *Cyp4f39* deficiency causes loss of type 2ω wax diesters in meibum lipids

WEs are molecules in which FA(s) and FAl(s) are ester-bonded (*Figure 5A*). Among WEs, those containing one and two ester bonds are called wax monoesters (referred to as WEs here) and wax diesters (WdiEs), respectively. WdiEs are further classified into type 1 and type 2 (*Nikkari, 1974*): in type 1 WdiEs, the backbone hydroxy FA is esterified with a FA and a FAl, whereas in type 2 WdiEs, the backbone fatty diol forms ester bonds with two FAs. In addition, both type 1 and 2 WdiEs have positional isomers: those with ester bonds at the C1 and ω positions of the backbone are type 1/2ω WdiEs, and those with these bonds at the C1 and α positions are type 1/2α WdiEs. Although the presence of type 1 and 2 WdiEs in meibum lipids has been suggested (*Chen et al., 2013*; *Butovich, 2017*), their ester bond positions (α or ω) have not been determined.

To determine the ester bond positions of type 2 WdiEs in meibum lipids and to quantify their levels, standards are required. As neither type 2ω nor type 2α WdiE standards are commercially available, we chemically synthesized type 2ω WdiE ([1,ω-*O*-C18:1]-C16:0) and type 2α WdiE ([1,α-*O*-C18:1]-C16:0) (*Figure 5—figure supplement 1*). Product ion scanning revealed that these WdiEs produce characteristic fragment ion(s): (M + H−[C18:1 FA−OH])$^+$ (*m/z* = 523.25) and (M + H−[C18:1 FA−OH]−$H_2O$)$^+$ (*m/z* = 505.20) from (1,ω-*O*-C18:1)-C16:0 WdiE; and (M + H−[C18:1 FA−OH]−$H_2O$)$^+$ (*m/z* = 505.20) from (1,α-*O*-C18:1)-C16:0 WdiE (*Figure 5B,C* and *Figure 5—figure*

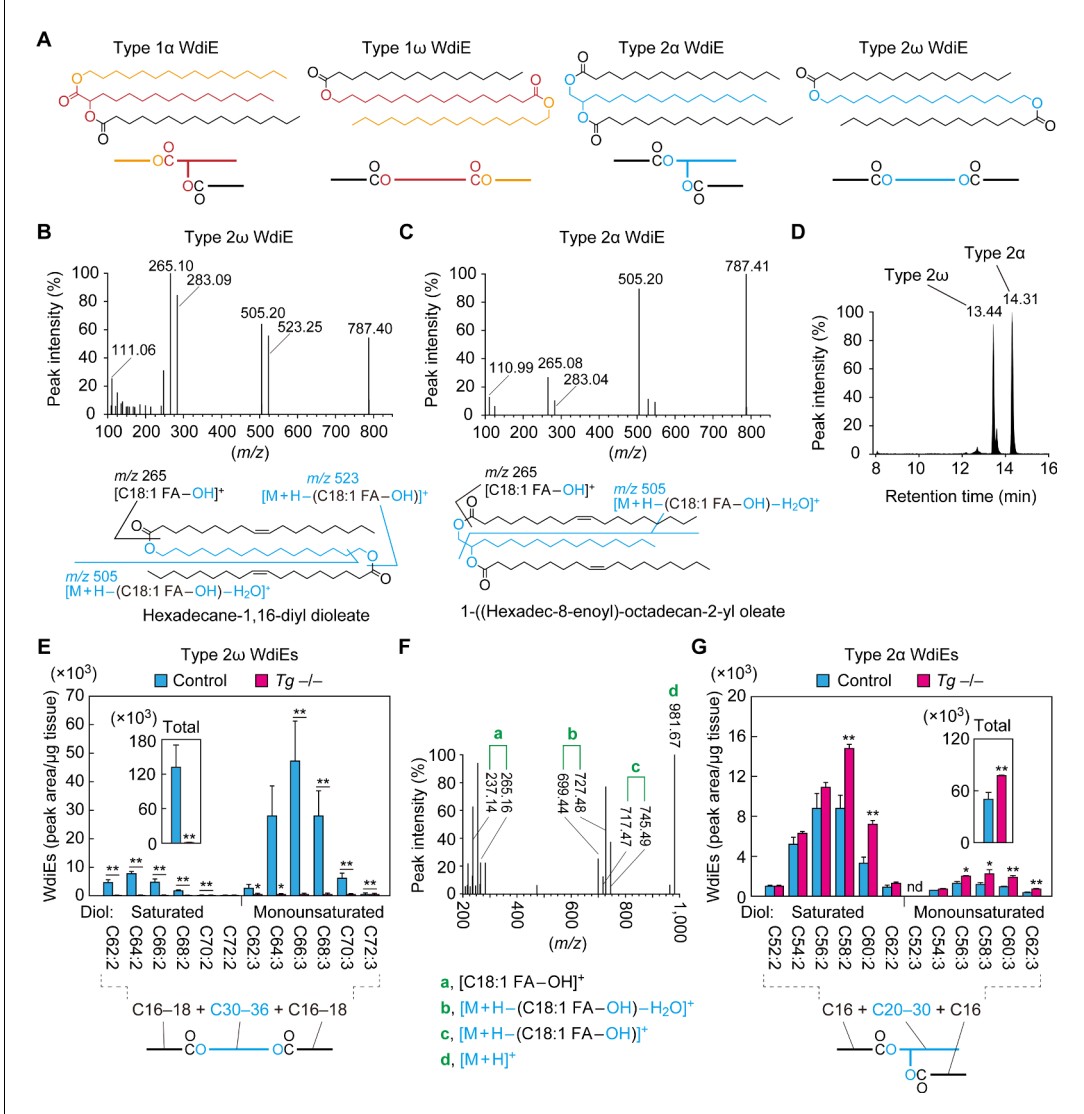

**Figure 5.** Absence of type 2ω WdiEs in meibomian glands from *Cyp4f39*-deficient mice. (**A**) Structures and simplified structures of type 1α, 1ω, 2α and 2ω WdiEs. Blue, red, and orange represent fatty diols, hydroxy FAs, and FAIs, respectively. (**B–D**) Product ion scanning of the type 2ω WdiE ([1,ω-*O*-C18:1]-C16:0) (**B**) and the type 2α WdiE [(1,α-*O*-C18:1)-C16:0] (**C**) was performed using LC-MS/MS by selecting [M + H]⁺ ions with *m/z* 787.7 as the precursor ions. The MS spectra (**B**, **C**), the predicted cleavage positions (**B**, **C**), and the LC chromatogram (**D**) are shown. Synthesis schemes for the type 2ω WdiE hexadecane-1,16-diyl dioleate ([1,ω-*O*-C18:1]-C16:0) and type 2α WdiE 1-([hexadec-8-enoyl]oxy)-octadecan-2-yl oleate ([1,α-*O*-C18:1]-C16:0), and their fragment predictions in MS product ion scanning are provided in *Figure 5—figure supplements 1–3*. (**E–G**) Lipids were extracted from the meibomian glands of 12-month-old control (*Tg-Cyp4f39*⁺/⁺ [n = 2] and *Tg-Cyp4f39*⁺/⁻ [n = 1]) (**E–G**) and *Tg-Cyp4f39*⁻/⁻ (n = 3) mice (**E**, **G**), and type 2ω WdiEs (**E**, **F**) and type 2α WdiEs (**G**) were analyzed by LC-MS/MS. (**E**, **G**) The peak areas of WdiE species of different chain lengths, with or without saturation, and their total amounts (insets) are shown. Values presented are means ± SD (*, p<0.05; **, p<0.01; Student's *t*-test). The simplified structure representing each WdiE is shown below the graph. nd, not detected; *Tg* –/–, *Tg-Cyp4f39*⁻/⁻. (**F**) Product ion scanning was performed by selecting the [C66:3 WdiE + H]⁺ ion, which has an *m/z* value of 981.7, as the precursor ion. The MS spectrum of type 2ω C66:3 WdiE is shown.

The online version of this article includes the following figure supplement(s) for figure 5:

**Figure supplement 1.** Synthesis schemes for type 2ω WdiE ([1,ω-*O*-C18:1]-C16:0) and type 2α WdiE ([1,α-*O*-C18:1]-C16:0).

**Figure supplement 2.** Predicted fragments from type 2ω WdiE.

**Figure supplement 3.** Predicted fragments from type 2α WdiE.

*supplements 2* and *3*). The retention time of (1,ω-*O*-C18:1)-C16:0 WdiE in the LC column was 13.44 min, whereas that of (1,α-*O*-C18:1)-C16:0 WdiE was 14.31 min (*Figure 5D*). This difference in retention time (0.87 min) indicates that type 2ω and type 2α WdiEs with the same molecular weight can be separated by LC.

On the basis of the above observation, type 2ω and type 2α WdiEs in meibum lipids prepared from control and *Tg-Cyp4f39⁻ᐟ⁻* mice were analyzed by LC-MS/MS in MRM mode, by setting (M + H−[C18:1 FA−OH]−H$_2$O)$^+$ as the fragment ions. We revealed the existence of type 2ω WdiEs in control mice (*Figure 5E*). The total carbon chain length of these species was 62–72, and the total number of double bonds was 2 or 3. The most abundant type 2ω WdiE species was C66:3 WdiE. Product ion scanning was conducted to reveal the molecular species of the FAs and fatty diols constituting it. Three fragment ions related to each C16:1 FA and C18:1 FA (C16:1 FA, *m/z* = 237.14, 727.48, and 745.49; C18:1 FA, *m/z* = 265.16, 699.44, and 717.47) were detected (*Figure 5F*). The ion strengths of the fragment ions related to C16:1 FA were all ~4 times higher than those of the corresponding fragment ions related to C18:1 FA, indicating that C16:1 FA and C18:1 FA comprised 78% and 22% of the FA species in C66:3 type 2ω WdiEs, respectively (*Table 1*). This means that C66:3 type 2ω WdiEs is a mixture of C16:1 FA/C34:1 fatty diol/C16:1 FA, C16:1 FA/C32:1 fatty diol/ C18:1 FA, and C18:1 FA/C30:1 fatty diol/C18:1 FA. Of these, the C16:1 FA/C34:1 fatty diol/C16:1 FA species was the most abundant (*Table 1*). Similar analyses were also conducted on other type 2ω WdiE species. In summary, type 2ω WdiEs contained a C30–C36 fatty diol (14% saturated and 86% monounsaturated) and two FAs (mainly C16:1 FA, with a small proportion of C18:1 FA) (*Figure 5E* and *Table 1*). In *Tg-Cyp4f39⁻ᐟ⁻* mice, the type 2ω WdiEs were greatly reduced relative to control

**Table 1.** FA composition of type 2ω/α WdiEs in meibum lipids and the most abundant molecular species predicted.

| WdiE type | Total carbon chain length and degree of unsaturation | FA composition (%) | | The most abundant molecular species predicted | |
| --- | --- | --- | --- | --- | --- |
| | | C16:1 FA | C18:1 FA | ω/α-OH diol | FAs |
| 2ω | C62:2 | 95 | 5 | C30:0 | C16:1–C16:1 |
| 2ω | C64:2 | 78 | 22 | C32:0 | C16:1–C16:1 |
| 2ω | C66:2 | 55 | 45 | C32:0 | C16:1–C18:1 |
| 2ω | C68:2 | 14 | 86 | C32:0 | C18:1–C18:1 |
| 2ω | C70:2 | 25 | 75 | C34:0 | C18:1–C18:1 |
| 2ω | C72:2 | 93 | 7 | C36:0 | C16:1–C20:1 |
| 2ω | C62:3 | 99 | 1 | C30:1 | C16:1–C16:1 |
| 2ω | C64:3 | 96 | 4 | C32:1 | C16:1–C16:1 |
| 2ω | C66:3 | 78 | 22 | C34:1 | C16:1–C16:1 |
| 2ω | C68:3 | 44 | 56 | C34:1 | C16:1–C18:1 |
| 2ω | C70:3 | 7 | 93 | C34:1 | C18:1–C18:1 |
| 2ω | C72:3 | 14 | 86 | C36:1 | C18:1–C18:1 |
| 2α | C52:2 | 98 | 2 | C20:0 | C16:1–C16:1 |
| 2α | C54:2 | 68 | 32 | C22:0 | C16:1–C16:1 |
| 2α | C56:2 | 91 | 9 | C24:0 | C16:1–C16:1 |
| 2α | C58:2 | 90 | 10 | C26:0 | C16:1–C16:1 |
| 2α | C60:2 | 70 | 30 | C28:0 | C16:1–C16:1 |
| 2α | C62:2 | 69 | 31 | C30:0 | C16:1–C16:1 |
| 2α | C54:3 | 90 | 10 | C22:1 | C16:1–C16:1 |
| 2α | C56:3 | 73 | 27 | C24:1 | C16:1–C16:1 |
| 2α | C58:3 | 98 | 2 | C26:1 | C16:1–C16:1 |
| 2α | C60:3 | 93 | 7 | C28:1 | C16:1–C16:1 |
| 2α | C62:3 | 83 | 17 | C30:1 | C16:1–C16:1 |

mice, regardless of chain length, and the total amount was 0.4% of that in control mice (**Figure 5E**). These results indicate that Cyp4f39 is essential for type 2ω WdiE production.

In addition to type 2ω WdiEs, type 2α WdiEs were also detected in control mice (**Figure 5G**). Type 2α WdiEs have a C20–C30 fatty diol (87% saturated and 13% monounsaturated) and two FAs (almost exclusively C16:1) (**Figure 5G** and **Table 1**). In *Tg-Cyp4f39*$^{-/-}$ mice, the total amount of type 2α WdiEs was approximately 1.5-fold that in control mice, with larger increases in the type 2α WdiE species with a chain length of ≥C56 (total carbon number) (**Figure 5G**). Therefore, Cyp4f39 is not involved in type 2α WdiE synthesis. This result is reasonable, considering that the substrates of Cyp4f39 are ≥C30 FAs and that the chain length of the fatty diol moiety of type 2α WdiEs is ≤C30.

### Loss of type 1ω WdiEs and Chl-OAHFAs in meibum lipids due to *Cyp4f39* deficiency

In the product ion scanning of C66:3 type 2ω WdiEs, we detected another peak (retention time, 16.72 min) in addition to the peak for type 2ω WdiEs (17.70 min) in the LC (**Figure 6A**). When this peak was subjected to fragment ion analysis, in addition to the precursor ion (*m/z* = 981.68), two product ions, with *m/z* values of 237.14 and 741.46, were detected (**Figure 6B**). The former *m/z* value corresponds to the ion derived from C16:1 FA ([C16:1 FA–OH]$^+$). The latter value matches the molecular weight of the precursor ion minus that of C16:1 FAl, suggesting that this ion is [M + H– C16:1 FAl]$^+$. This suggests that the unknown peak corresponds to type 1 WdiE, consisting of C16:1 FA, ω- or α-OH C34:1 FA, and C16:1 FAl (**Figure 6C**).

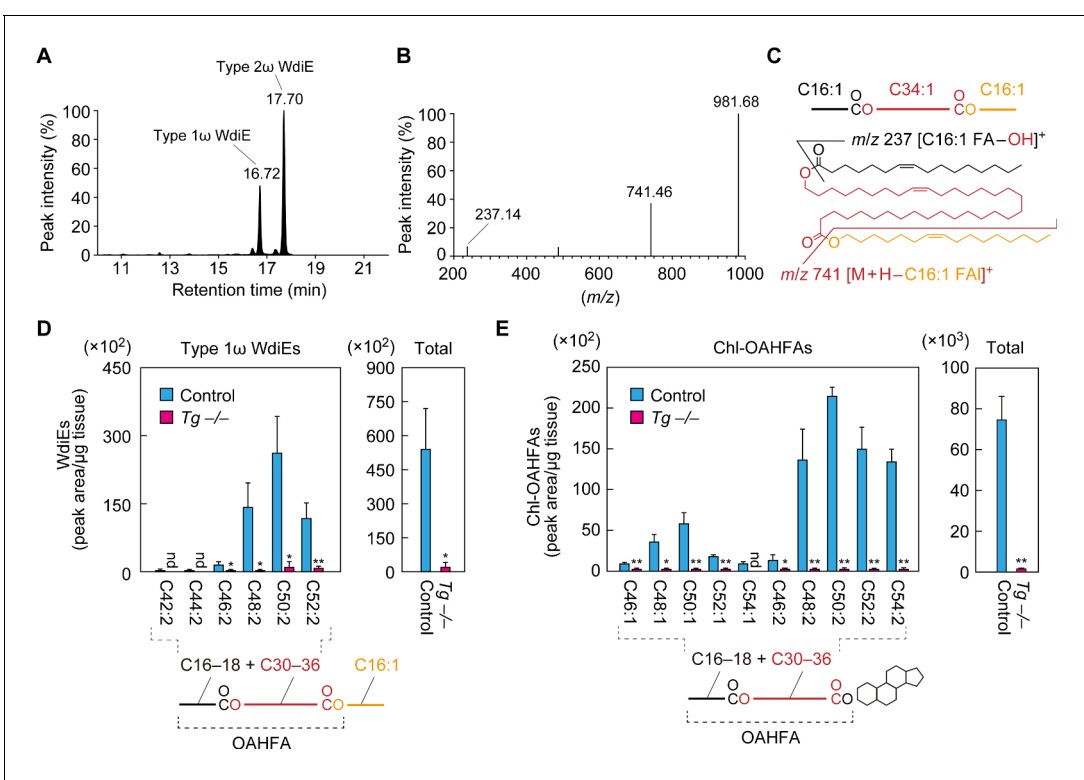

**Figure 6.** Absence of type 1ω WdiEs and Chl-OAHFAs in meibomian glands from *Cyp4f39*-deficient mice. (**A–C**) Meibum lipids were prepared from 12-month-old *Tg-Cyp4f39*$^{+/+}$ mice and subjected to product ion scanning by selecting the [C66:3 WdiE + H]$^+$ ion, with an *m/z* value of 981.7, as the precursor ion. LC chromatogram of C66:3 WdiEs (**A**) and MS spectrum of type 1ω C66:3 WdiE (**B**) are shown. The structure of the type 1ω C66:3 WdiE is illustrated (**C**), as predicted from the fragment ion analysis in panel (**B**). (**D, E**) Lipids were extracted from the meibomian glands of 12-month-old control (*Tg-Cyp4f39*$^{+/+}$ [n = 2] and *Tg-Cyp4f39*$^{+/-}$ [n = 1]) and *Tg-Cyp4f39*$^{-/-}$ (n = 3) mice. Then, type 1ω WdiEs (**D**) and Chl-OAHFAs (**E**) were analyzed by LC-MS/MS. The peak areas (left-hand panels) of lipid species with different chain lengths, with or without saturation, and their total amounts (right-hand panels) are shown. Values presented are means ± SD (*, p<0.05; **, p<0.01; Student's *t*-test). The simplified structure of each lipid is shown below the graphs. nd, not detected; *Tg* –/–, *Tg-Cyp4f39*$^{-/-}$.

We then analyzed the type 1 WdiEs in meibum lipids by LC-MS/MS in MRM mode, setting (M + H–C16:1 FAl)$^+$ as the fragment ion to detect. In control mice, type 1 WdiEs containing C16:1 FAl, C16:1 or C18:1 FA, and C30–36 hydroxylated FAs (the sum of the carbon chain lengths of the FA and hydroxy FA moieties is C42–C52) were detected (*Figure 6D*). In *Tg-Cyp4f39*$^{-/-}$ mice, all of the type 1 WdiE species detected in the control mice were greatly reduced, and the total amount of type 1 WdiEs was ~4% of that in control mice. On the basis of this result, combined with the fact that Cyp4f39 is a FA ω-hydroxylase, we concluded that the type 1 WdiEs detected in the control mice were type 1ω WdiEs.

The existence of Chl-OAHFAs in meibum lipids has previously been reported (*Chen et al., 2013*; *Butovich, 2017*). To examine whether Cyp4f39 is involved in the production of these species, we analyzed Chl-OAHFA levels in meibum lipids from control and *Tg-Cyp4f39*$^{-/-}$ mice by LC-MS/MS. Chl-OAHFAs with C46–C54 in the OAHFA moiety were detected in control mice, but they were almost absent in *Tg-Cyp4f39*$^{-/-}$ mice (*Figure 6E*). As Cyp4f39 is involved in the production of OAH-FAs composed of a C30–C36 ω-OH FA and a C16:1 FA (*Figure 4C*), the majority of the Chl-OAHFAs detected in the control mice may also contain a C30–C36 ω-OH FA and a C16:1 FA (although they also contain a small fraction of C18:1 FA). In summary, Cyp4f39 is involved not only in the production of OAHFAs and type 2ω WdiEs, but also in that of type 1ω WdiEs and Chl-OAHFAs.

## Increases in CE and WE levels in meibomian glands of *Cyp4f39*-deficient mice

To examine the effects of *Cyp4f39* deficiency on meibum lipids other than OAHFAs and their derivatives (type 1/2ω WdiEs and Chl-OAHFAs), meibum lipids prepared from *Tg-Cyp4f39*$^{+/+}$ and *Tg-Cyp4f39*$^{-/-}$ mice were separated by normal-phase thin-layer chromatography (TLC), in which the mobility of lipids with lower polarities is greater. We used two resolving solutions for the TLC: one suitable for separating total meibum lipids, and one for the separation of CEs and WEs. CEs and WEs are the least polar of the meibum lipids, and they were detected in the uppermost two bands on the TLC plates (*Figure 7A*). We confirmed the reduction of OAHFAs and their derivatives in *Tg-Cyp4f39*$^{-/-}$ mice (*Figure 7A*). Judging from their mobility in the TLC, WdiEs and Chl-OAHFAs showed similar degrees of polarity, and these were intermediate between the polarities of CEs/WEs and OAHFAs.

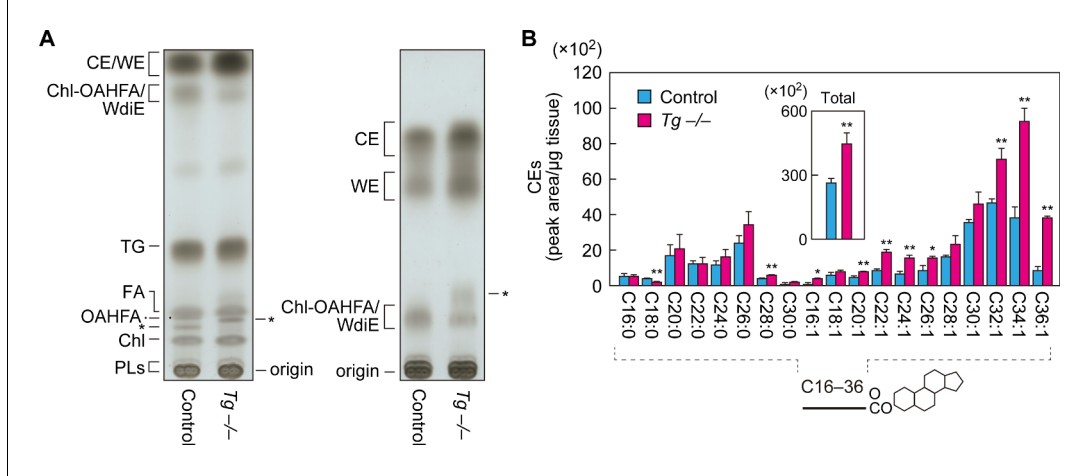

**Figure 7.** Increased CEs and WEs in *Cyp4f39*-deficient mice. (**A**) Meibum lipids prepared from 12-month-old control (*Tg-Cyp4f39*$^{+/+}$) and *Tg-Cyp4f39*$^{-/-}$ mice were separated by TLC using solution 1 (hexane/diethyl ether/acetic acid [90:25:1, v/v]; left) or solution 2 (hexane/toluene [1:1, v/v; right]) as the resolving solution, followed by detection using copper phosphate reagent. Asterisks indicate unknown lipids. (**B**) Lipids were extracted from the meibomian glands of 12-month-old control (*Tg-Cyp4f39*$^{+/+}$ [n = 2] and *Tg-Cyp4f39*$^{+/-}$ [n = 1]) and *Tg-Cyp4f39*$^{-/-}$ (n = 3) mice, and CEs were analyzed by LC-MS/MS. The peak areas for CE species with different FA chain lengths, with or without saturation, and their total amounts (inset) are shown. Values presented are means ± SD (*, p<0.05; **, p<0.01; Student's *t*-test). The simplified structure of CE is shown below the graph. Chl, cholesterol; TG, triglyceride; PL, phospholipid; *Tg –/–*, *Tg-Cyp4f39*$^{-/-}$.

Both CEs and WEs were more abundant in *Tg-Cyp4f39*<sup>−/−</sup> mice than in *Tg-Cyp4f39*<sup>+/+</sup> mice (*Figure 7A*). Next, we quantified CE levels by LC-MS/MS. CEs containing C16–C30 saturated or C16–C36 monounsaturated FAs were detected in the meibum lipids of control mice (*Figure 7B*). In *Tg-Cyp4f39*<sup>−/−</sup> mice, CEs of many chain lengths were more abundant. In particular, the levels of ≥C32 species were higher in *Tg-Cyp4f39*<sup>−/−</sup> mice than in the control mice: the total amount was about 1.7 times higher. These results suggest that ≥C32 FAs that were not ω-hydroxylated by Cyp4f39 were used for the synthesis of CEs.

## Discussion

CEs and WEs, which are the most abundant components of the TFLL, form a nonpolar lipid sublayer that is in contact with the external environment. OAHFAs constitute an amphiphilic lipid sublayer and are thought to have a role in stabilizing the tear film by creating an interface between the non-polar lipid sublayer and the aqueous layer (*Butovich et al., 2009*; *King-Smith et al., 2013*). However, it has not yet been possible to create a model organism that lacks OAHFAs, due to a lack of knowledge about their biosynthesis pathway. Therefore, the actual function of OAHFAs has not yet been elucidated. In the present study, we found that FA ω-hydroxylase Cyp4f39 is involved in OAHFA production (*Figure 4*). *Cyp4f39*-deficient mice exhibited dry eye accompanied by meibomian gland obstruction and tear film destabilization (*Figure 2*). In these mice, not only OAHFAs but also their derivatives (type 1ω WdiEs, type 2ω WdiEs, and Chl-OAHFAs) were reduced (*Figures 5* and *6*). These results suggest that the observed dry eye was caused by the combined effects of decreased production of OAHFAs and their derivatives.

Cyp4f39 and its human ortholog, CYP4F22, show high ω-hydroxylase activity toward ≥C30 FAs (*Ohno et al., 2015*; *Miyamoto et al., 2020*). Alkyl chains are classified according to their chain length: those with C11–C20 are long-chain, whereas those with ≥C21 are very-long-chain (VLC). Long-chain acyl-CoAs are converted to VLC acyl-CoAs through the FA elongation cycle, which comprises four reactions: condensation, reduction, dehydration, and reduction (*Kihara, 2012*). The condensation reaction, which is the rate-limiting step of the FA elongation cycle, is catalyzed by FA elongases. Mammals have seven FA elongase isozymes (ELOVL1–7) that have different substrate specificities (*Kihara, 2012*). It is likely that ELOVL1, ELOVL3, and ELOVL4 are involved in the production of ≥C30 monounsaturated very long chain fatty acids (VLCFAs), the components of OAHFAs. ELOVL1 and ELOVL3 are active toward saturated and monounsaturated C22–C24 acyl-CoAs (*Ohno et al., 2010*; *Sassa et al., 2018*), whereas ELOVL4 shows activity toward ≥C24 acyl-CoAs (*Vasireddy et al., 2007*; *Agbaga et al., 2008*). The contribution of ELOVL1 to FA elongation is high for saturated VLC acyl-CoAs but low for monounsaturated VLC acyl-CoAs (*Sassa et al., 2013*; *Sassa et al., 2018*). Since most of OAHFAs are composed of monounsaturated VLCFAs (*Figure 4*), OAHFAs are not decreased in *Elovl1*-deficient mice, in which the expression of *Elovl3* is increased as a compensatory mechanism (*Sassa et al., 2018*). However, CEs and WEs, which are rich in saturated VLCFAs and VLCFAls, respectively, are shortened in *Elovl1*-deficient mice, leading to a dry eye-like phenotype (*Sassa et al., 2018*). In addition, plugging at the meibomian gland orifice was observed in *Elovl4* mutant mice (*McMahon et al., 2014*).

We predict that the OAHFA biosynthesis pathway in the meibomian gland is as follows. First, long-chain acyl-CoAs are elongated by ELOVL1 and ELOVL3 to generate VLC acyl-CoAs, and then they are further elongated to ≥C30 acyl-CoAs by ELOVL4 (*Figure 8A*). After the removal of CoA, the resulting ≥C30 VLCFAs are ω-hydroxylated by CYP4F22 (Cyp4f39 in mice). Finally, an C16:1/C18:1 acyl group is transferred from C16:1/C18:1-CoA to the ω-OH VLCFAs, generating OAHFAs. In these reactions, the thioesterase that catalyzes the CoA removal and the acyltransferase that catalyzes the final ester bond formation have not yet been identified.

Although the existence of type 2α/ω WdiEs in meibum lipids has been suggested (*Chen et al., 2013*; *Butovich, 2017*), it has not been determined whether they are α or ω positional isomers. In the present study, we found that both type 2ω and type 2α WdiEs are present, and that type 2ω WdiEs are synthesized in a Cyp4f39-dependent manner (*Figure 5*). Furthermore, we revealed that Cyp4f39 is also involved in the production of type 1ω WdiEs and Chl-OAHFAs (*Figure 6*). We propose that the synthesis pathway for these OAHFA derivatives is as follows (*Figure 8A*). First, OAHFAs are converted to (*O*-acyl)-ω-OH (OAH) acyl-CoAs. Next, OAH acyl-CoAs are transferred to FAls

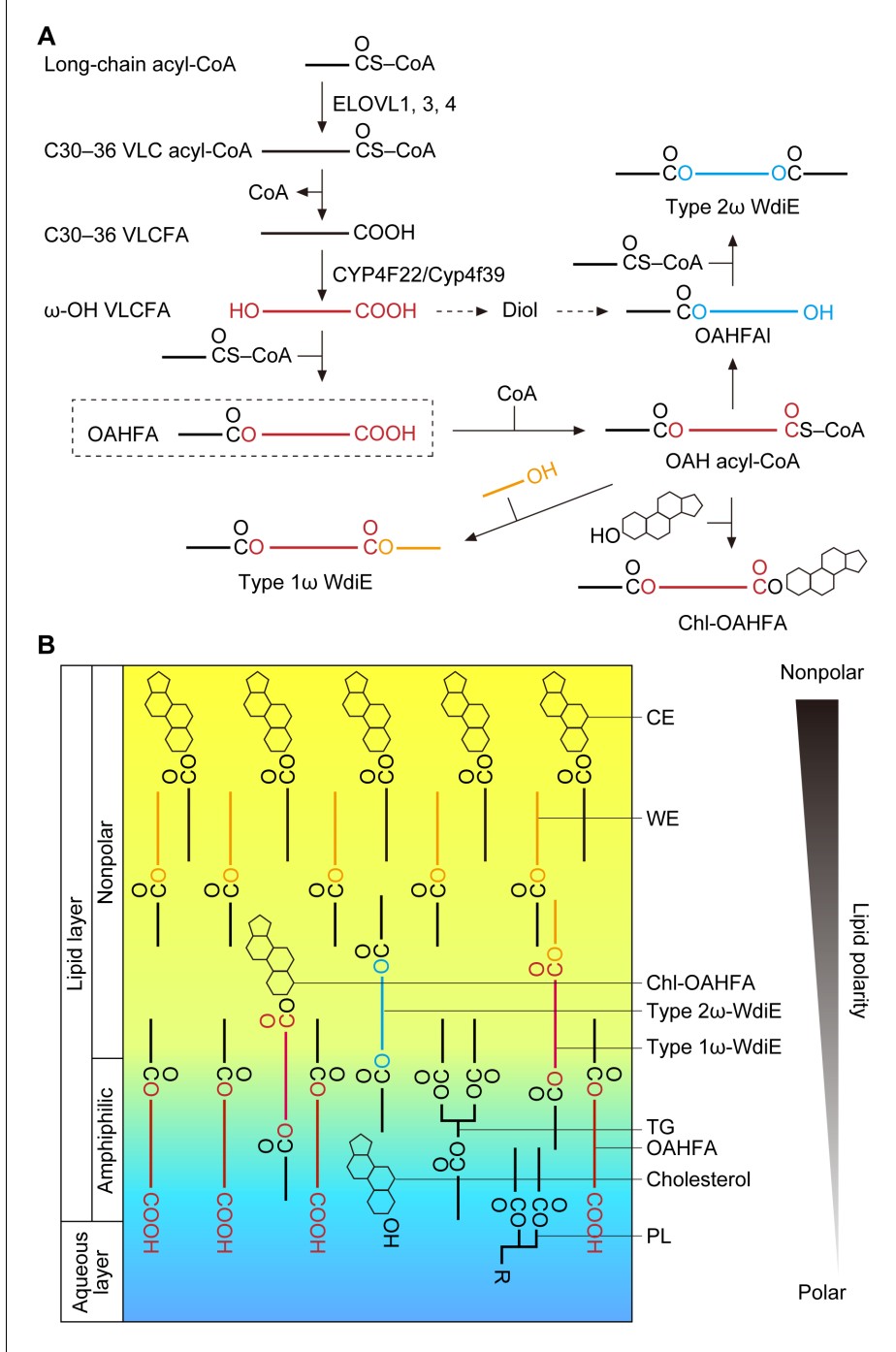

**Figure 8.** Models of the synthesis pathways of OAHFAs and their derivatives and of the lipid polarity gradient in TFLL. (**A**) Model of synthesis pathways of OAHFAs and their derivatives. Meibum lipids are shown as simplified structures (red, ω-OH FA; orange, FAl; blue, fatty diol). (**B**) Model of lipid polarity gradient formation by meibum lipids in the TFLL. CEs and WEs, which have the lowest polarity among meibum lipids, form the nonpolar lipid sublayer that faces the external environment. OAHFAs, phospholipids (PLs), cholesterol, and triglycerides (TGs) form the amphiphilic lipid sublayer that contacts the aqueous layer. Chl-OAHFAs and WdiEs are located at the interface between these two sublayers and have a role in connecting them.

or cholesterol, producing type 1ω-WdiEs or Chl-OAHFAs, respectively. Alternatively, OAH acyl-CoAs are converted to OAHFAls by acyl-CoA reductase, followed by ester bond formation with FAs, generating type 2ω WdiEs. However, another pathway is also possible for type 2ω WdiE synthesis, in which ω-OH VLCFAs are reduced to α,ω- fatty diols, followed by conversion to OAHFAls and then to type 2ω WdiEs.

At present, the acyltransferase involved in C16:1 OAHFA production (i.e., forming the ester bond between a C16:1 FA and an ω-OH VLCFA) is unknown. However, it is possible that C18:2 OAHFA, which was detected at higher levels than C16:1 OAHFA in our meibum lipid preparation (*Figure 4*), is derived from ω-*O*-acylceramides via degradation. ω-*O*-Acylceramides, which are important for the skin barrier and exist specifically in the epidermis (*Kihara, 2016*), consist of a long-chain base, an ω-OH FA (C30–C36), and linoleic acid (C18:2); the conjugate of the latter two components is C18:2 OAHFA. The transacylase PNPLA1 catalyzes the formation of an ester bond between ω-OH FA and linoleic acid using triglyceride as the linoleic acid donor (*Ohno et al., 2017*). We speculate that the C18:2 OAHFA that was detected in meibum lipids is derived from the keratinized epithelial cells constituting the meibomian gland ducts or from epidermis contamination in the samples. On the other hand, since PNPLA1 can produce few ω-*O*-acylceramides containing C16:1 FA (*Hirabayashi et al., 2019*), it is unlikely that it is involved in the production of C16:1 OAHFA in meibocytes.

Tears are secreted when the eyes are closed, and the formation of the tear film begins when the eyes open and involves two processes (*Brown and Dervichian, 1969*; *Willcox et al., 2017*). First, the liquid components of the aqueous layer that has accumulated on the lower eyelid side extend upward due to the negative pressure in the upper eyelid rim that is generated when the eyes are opened. At that time, the TFLL lipids (meibum lipids) remain on the lower eyelid side but are hardly present on the upper eyelid side, creating a surface pressure gradient. Subsequently, the TFLL lipids spread on the aqueous layer from the lower eyelid side to the upper eyelid side, cancelling out the surface pressure gradient (this is known as the Gibbs–Marangoni effect) (*Brown and Dervichian, 1969*; *King-Smith et al., 2009*). For the TFLL to extend, the TFLL and aqueous layers must interact with each other. However, CEs and WEs, which constitute most of the TFLL, are only weakly polar and can barely interact with the aqueous layer, whereas the polar phospholipids are not able to interact sufficiently with the nonpolar CEs and WEs. Therefore, OAHFAs, which have a polar group as well as a long hydrophobic chain, may play an important role in connecting the aqueous layer to the nonpolar lipid sublayer composed of CEs and WEs. Furthermore, in the present study, we have revealed that type 1ω WdiEs, type 2ω WdiEs, and Chl-OAHFAs, all derived from OAHFAs, have polarities intermediate between those of OAHFAs and CEs/WEs. This suggests that these OAHFA derivatives are located between OAHFAs and CEs/WEs and function to connect them. Therefore, we speculate that a lipid polarity gradient is formed from the aqueous layer/TFLL boundary to the external environment (from phospholipids, to OAHFAs, then OAHFA derivatives, and finally CEs/WEs), and that this gradient keeps the TFLL stably on top of the aqueous layer (*Figure 8B*). In fact, *Tg-Cyp4f39*$^{-/-}$ mice showed tear accumulation on the lower eyelid side (*Figure 1F*), which indicates increased surface tension and decreased interaction between the aqueous layer and the TFLL. In addition, these mice exhibited plugging at the orifices of the meibomian glands (*Figure 2D*). It is likely that the decreased interaction between the aqueous layer and the TFLL causes accumulation of meibum lipids at the eyelid rim, leading to abnormal secretion of meibum lipids from the meibomian glands. The plugging observed in *Tg-Cyp4f39*$^{-/-}$ mice was mild (semi-liquid state), and their meibomian glands were not enlarged. If secretion of meibum lipids had been strongly inhibited, the meibomian glands would have been expected to swell. Therefore, we speculate that that most meibum lipids are secreted in *Tg-Cyp4f39*$^{-/-}$ mice. However, we cannot exclude the possibility that the slight reduction in the secretion of total meibum lipids (not only OAHFA and its derivatives, but also other meibum lipids) due to the plugging causes further progression of dry eye in *Tg-Cyp4f39*$^{-/-}$ mice.

In the present study, we have revealed that Cyp4f39/CYP4F22 is involved in the production of OAHFAs and their derivatives, and have obtained some clues concerning their biosynthesis pathways. Furthermore, on the basis of these findings, we propose a model in which the lipid polarity gradient in the TFLL is important for the stabilization of the TFLL on top of the aqueous layer. Although dry eye drugs targeting the aqueous and glycocalyx layers exist, there are no drugs targeting the TFLL. Considering that the majority of dry eye disease is caused by abnormalities in the

TFLL, future studies are needed to develop drugs that target this layer, such as eye drops containing OAHFAs/OAHFA derivatives or drugs that promote their production.

# Materials and methods

## Key resources table

| Reagent type (species) or resource | Designation | Source or reference | Identifiers | Additional information |
|---|---|---|---|---|
| Mouse: M. musculus (C57BL/6J) | C57BL/6J; wildtype | Japan SLC | | |
| Mouse: M. musculus (C57BL/6J) | Cyp4f39+/− | Miyamoto et al., 2020 | | |
| Mouse: M. musculus (C57BL/6J) | Tg-Cyp4f39−/− | This study | | |
| Antibody | Mouse monoclonal anti-FLAG M2 | Merck | Cat# 3165 RRID:AB_259529 | (1 μg/mL) |
| Antibody | Mouse monoclonal anti-GAPDH 5A12 | FUJIFILM Wako Pure Chemical | Cat# 01625523 RRID:AB_2814991 | (1:2000) |
| Antibody | Anti-mouse IgG HRP-linked F(ab')₂ fragment | GE Healthcare Life Sciences | Cat# NA9310 RRID:AB_772193 | (1:7500) |
| Chemical compound, drug | Uranine (sodium fluorescein) | Tokyo Chemical Industry | Cat# F0096 | |
| Chemical compound, drug | Pentobarbital sodium salt | Tokyo Chemical Industry | Cat# P0776 | |
| Chemical compound, drug | Isoflurane | FUJIFILM Wako Pure Chemical | Cat# 099–06571 | |
| Chemical compound, drug | Super Fix | KURABO | Cat# KY-500 | |
| Chemical compound, drug | 22-Hydroxydocosanoic acid (ω-OH behenic acid) | Larodan | Cat# 14–2222 | |
| Chemical compound, drug | Triethylamine | FUJIFILM Wako Pure Chemical | Cat# 202–02646 | |
| Chemical compound, drug | Oleoyl chloride | FUJIFILM Wako Pure Chemical | Cat# 329–79572 | |
| Chemical compound, drug | 1,16-Hexadecanediol | Tokyo Chemical Industry | Cat# H0552 | |
| Chemical compound, drug | 1,2-Hexadecanediol | Tokyo Chemical Industry | Cat# H0993 | |
| Chemical compound, drug | 4-Dimethylaminopyridine | FUJIFILM Wako Pure Chemical | Cat# 040–19213 | |
| Commercial assay or kit | NucleoSpin RNA Kit | Machery-Nagel | Cat# U0955C | |
| Commercial assay or kit | One Step TB Green PrimeScript RT-PCR Kit II | Takara Bio | Cat# RR086A | |
| Commercial assay or kit | AMP+ MaxSpec Kit | Cayman Chemical | Cat# 710000 | |
| Recombinant DNA reagent | pCE-puro 3×FLAG-1 | Kihara et al., 2006 | | |
| Recombinant DNA reagent | pCE-puro 3×FLAG-Cyp4f39 | This study | | |
| Recombinant DNA reagent | pH3700-pL2 | Carroll et al., 1993 | | |

*Continued on next page*

*Continued*

| Reagent type (species) or resource | Designation | Source or reference | Identifiers | Additional information |
|---|---|---|---|---|
| Recombinant DNA reagent | pH3700-pL2−3×FLAG-Cyp4f39 | This study | | |
| Software, algorithm | MassLynx software | Waters | RRID:SCR_014271 | |
| Software, algorithm | Microsoft Excel software | Microsoft | RRID:SCR_016137 | |
| Software, algorithm | JMP13 software | SAS Institute | | |

## Mice

The *3×FLAG-Cyp4f39* transgene (*Tg-Cyp4f39*), under the control of the *IVL* promoter, was constructed as follows. *Cyp4f39* was amplified by PCR from cDNA from mouse testis and cloned into the *3×FLAG* vector pCE-puro 3×FLAG-1 (*Kihara et al., 2006*), producing the pCE-puro 3×FLAG-Cyp4f39 plasmid. The *3×FLAG-Cyp4f39* fragment in the pCE-puro 3×FLAG-Cyp4f39 plasmid was then transferred to the pH3700-pL2 vector containing a human *IVL* promoter (*Carroll et al., 1993*), generating the pH3700-pL2−3×FLAG-Cyp4f39 plasmid. The *IVL-Tg-Cyp4f39* fragment in the pH3700-pL2−3×FLAG-Cyp4f39 plasmid was excised, purified, and microinjected into fertilized eggs from C57BL/6J mice. Genomic DNA samples were prepared from the mice thus generated, and the presence of the *Tg-Cyp4f39* transgene was detected by genomic PCR using primers (Involucrin-Tg-F and Cyp4f39-R; *Supplementary file 1*) to amplify the transgene fragment (678 bp). The transgene was present in four founder mice, and their epidermis was subjected to immunoblotting with anti-FLAG antibody to validate the expression of 3×FLAG-Cyp4f39 protein. The transgenic mice expressing 3×FLAG-Cyp4f39 protein (*Tg-Cyp4f39*[+/+] mice) were crossed with *Cyp4f39*[+/−] mice (*Miyamoto et al., 2020*) to produce *Tg-Cyp4f39*[+/−] mice. This strain was maintained by repeated back-crossing with C57BL/6J mice. *Tg-Cyp4f39*[−/−] mice were prepared by crossing *Tg-Cyp4f39*[+/−] mice. Genotyping for *Tg-Cyp4f39* was performed as described above, and that for *Cyp4f39* was conducted by genomic PCR using p1 and p2 primers as described previously (*Miyamoto et al., 2020*). The mice were housed in a specific pathogen-free, controlled environment (room temperature, 23 ± 1°C; humidity 50 ± 5%; 12 hr light/12 hr dark cycle). They were fed with normal diet (PicoLab Rodent Diet 20; LabDiet, St. Louis, MO) and given water ad libitum. All animal experiments were approved by the institutional animal care and use committee of Hokkaido University (Permit Number: 17–0017).

## Immunoblotting

Immunoblotting was performed as described previously (*Kitamura et al., 2017*), using anti-FLAG (1 µg/mL; Merck, Darmstadt, Germany) or anti-GAPDH antibody (1:2000 dilution; FUJIFILM Wako Pure Chemical, Osaka, Japan) as the primary antibody and anti-mouse IgG, HRP-linked F(ab')$_2$ fragment (1:7500 dilution; GE Healthcare Life Sciences, Little Chalfont, UK) as the secondary antibody. Chemiluminescence was performed using Western Lightning Plus-ECL (Thermo Fischer Scientific, Waltham, MA).

## BUT measurement and corneal damage scoring

Soon after 3 µL of 0.5% liquid sodium fluorescein (Tokyo Chemical Industry, Tokyo, Japan) had been loaded onto the mouse eyes, the mice were manually forced to blink three times, to ensure that the fluorescein solution covered the entire surface of the eyeballs. The mouse eye surfaces were then observed under a slit lamp microscope (Slit lamp RO800; Luneau Technology Operations, Pont-de-l'Arche, France) with a cobalt blue filter. BUT represents the elapsed time (in seconds) from the moment when the eyeball surface was covered with the fluorescein solution until the uniform staining was destroyed. Scoring of corneal epithelial damage was performed immediately after measurement of BUT. Mice were anesthetized by intraperitoneal injection of 0.05 mg/g (body weight) pentobarbital (Tokyo Chemical Industry), and then their eye surfaces were observed under a slit lamp microscope with cobalt blue and yellow filters. Scoring of corneal damage was performed

according to a previous report (*Lemp, 1995*). The eyeball surface was divided into five sections, and the corneal damage in each section was evaluated as grade 0 to 3. The corneal damage score was calculated as the sum of the section scores. The grades were determined on the basis of the fraction of the area over which punctate staining was observed: 0, <1/3 area; 1, 1/3 to <2/3 area; 2, 2/3 to <3/3 area; 3, entire area or presence of filamentary keratitis.

### Tear quantity measurement

After the mice had been anesthetized through inhalation of isoflurane (FUJIFILM Wako Pure Chemical), tear quantities were measured using the phenol red-thread test using Zone-Quick (Showa Yakuhin Kako, Tokyo, Japan), according to the manufacturer's manual.

### Hematoxylin and eosin staining

The eyeballs and eyelids were fixed with Super Fix (Kurabo, Osaka, Japan) at 4°C for ≥24 hr. Preparation of paraffin sections and staining with hematoxylin and eosin were performed as described previously (*Sassa et al., 2013*). Brightfield images were observed under a Leica DM5000 B microscope (Leica Microsystems, Wetzlar, Germany).

### Real-time quantitative RT-PCR

Meibomian glands were removed from the upper and lower eyelids of mice under a stereomicroscope (Stemi DV4; Carl Zeiss, Oberkochen, Germany), and total RNA samples were prepared using a NucleoSpin RNA Kit (Machery-Nagel, Dueren, Germany), according to the manufacturer's instructions. Real-time quantitative RT-PCR was performed using 50 ng/μL of total RNAs, primer pairs (Awat1-F/-R, Awat2-F/-R, Far1-F/-R, Far2-F/-R, Soat1-F/-R, Cyp4f39-F/-R2, and Hprt-F/-R; *Supplementary file 1*), and One Step TB Green PrimeScript RT-PCR Kit II (Takara Bio, Shiga, Japan), as described previously (*Miyamoto et al., 2020*).

### Lipid analyses by LC-MS/MS

Meibomian glands were removed from the upper and lower eyelids of the mice under a stereomicroscope and subjected to lipid extraction as follows. Meibomian glands (2.0–3.8 mg) were transferred to zirconia-bead-containing tubes (SARSTEDT, Nümbrecht, Germany) and suspended in 450 μL of chloroform/methanol/12 M formic acid (100:200:1, v/v). Samples were vigorously mixed (4°C, 5000 rpm, 2 min) using a Micro Smash MS-100 (TOMY Seiko, Tokyo, Japan). After removing the zirconia beads, the samples were mixed with 150 μL of chloroform and 270 μL of water. Phase separation was performed by centrifugation (20,400 × g, room temperature, 5 min), and the organic phase was recovered. The aqueous phase was again subjected to lipid extraction by mixing with 150 μL of chloroform, followed by centrifugation. The organic phase was recovered and combined with the previous one. The extracted lipids were dried and dissolved in 200 μL of hexane. Next, the lipids were subjected to a second phase separation by mixing with 200 μL of water and centrifugation (20,400 × g, room temperature, 3 min). The organic phase was recovered, and the lipids were re-extracted from the aqueous phase by mixing with 200 μL of hexane, followed by centrifugation. The combined lipid extracts (meibum lipid fraction) were dried.

FAs were prepared from the meibum lipid fraction by alkaline treatment as described previously (*Jojima et al., 2020*). FAs and OAHFAs were derivatized with AMPP using an AMP+ MaxSpec Kit (Cayman Chemical, Ann Arbor, MI) as described previously (*Hancock et al., 2018*). Meibum lipids were dissolved in chloroform/methanol (1:2, v/v) at a concentration of 0.1 mg tissue/μL, and 0.1 μL (FAs) or 1.0 μL (OAHFAs) of each sample was subjected to derivatization, according to the manufacturer's manual. After the reaction, acetonitrile was added so that the sample volume became 100 μL.

LC-MS/MS analyses were performed using ultra-performance LC (UPLC) coupled with electrospray ionization tandem triple-quadrupole mass spectrometry (Xevo TQ-S; Waters, Milford, MA). Lipids were separated by UPLC using a reverse phase column (Acquity UPLC CSH C18 column; length, 100 mm; particle size, 1.7 μm; inner diameter, 2.1 mm; Waters) at 55°C. The amounts of meibum lipids injected onto the UPLC column were 0.5 μg (tissue weight) for AMPP-FAs, 5 μg for AMPP-OAHFAs, and 50 μg for each of type 2α/ω WdiEs, type 1α/ω WdiEs, Chl-OAHFAs, and CEs. Lipid separation by UPLC was performed at a flow rate of 0.3 mL/min using a gradient system in which

mobile phase A (acetonitrile/water [3:2, v/v] containing 5 mM ammonium formate) and mobile phase B (acetonitrile/2-propanol [1:9, v/v] containing 5 mM ammonium formate) were mixed. The gradient conditions for AMPP-OAHFA measurement were as follows: 0 min, 40% B; 0–18 min, gradient to 100% B; 18–23 min, 100% B; 23–30 min, gradient to 40% B. For the type 2α/ω WdiEs, type 1α/ω WdiEs, Chl-OAHFAs, and CEs, the gradient conditions were as follows: 0 min, 60% B; 0–21 min, gradient to 100% B; 21–25 min, 100% B; 25–30 min, gradient to 60% B. Ionization of lipids was performed by electrospray ionization. Quantitative analyses were performed by MRM in positive ion mode. The cone voltage was set at 35 V, and an appropriate collision energy was set for each lipid (AMPP-OAHFA, 60 eV; type 1α/ω and type 2α/ω WdiEs, 20 eV; Chl-OAHFAs and CEs, 15 eV). The $m/z$ values of the precursor and product ions of each lipid species were set in the mass filters Q1 and Q3, respectively (*Yamamoto et al., 2020*; *Supplementary files 2–6*). Product ion scanning was performed for fragment ion analyses. In the analyses of OAHFAs, $[M - H]^-$ was detected as the product in negative ion mode, but in the analyses of type 1 and 2 WdiEs, $[M + H]^+$ was detected in positive ion mode. Data analyses were performed using MassLynx software (Waters).

## Lipid analyses by TLC

The upper and lower eyelids were removed from the mice and cooled on ice. Meibomian gland contents were squeezed out from the orifices of the meibomian glands using two pairs of tweezers under a stereomicroscope, while keeping the tissue on ice as described previously (*Butovich et al., 2012*). The extracted contents were dissolved in chloroform/methanol (1:2, v/v) and dried. They were then dissolved in 450 μL of chloroform/methanol/12 M formic acid (100:200:1, v/v), mixed with 150 μL of chloroform and 270 μL of water, and centrifuged at 20,400 × g for 5 min at room temperature. Lipids were recovered from the organic phase, dried, suspended in chloroform/methanol (1:2, v/v), and separated using normal-phase TLC using high-performance TLC plates (Glass HPTLC Silica gel 60 plates; Merck) with solution 1 (hexane/diethyl ether/acetic acid [90:25:1, v/v]) or solution 2 (hexane/toluene [1:1, v/v]) as the resolving solution. Lipids were detected by spraying the plates with a copper phosphate reagent (3% $CuSO_4$[w/v] in 8% [v/v] aqueous phosphoric acid solution), followed by heating at 180°C.

## Chemical synthesis of OAHFA and type 2 WdiEs

The OAHFA (*O*-C18:1)-ω-OH C22:0 FA was synthesized as follows. 22-Hydroxy docosanoic acid (ω-OH C22:0 FA; 1.2 mg, 3.4 μmol; Larodan AB, Solna, Sweden) was dissolved in 400 μL of tetrahydrofuran, mixed with triethylamine (3.7 μL, 27.0 μmol; FUJIFILM Wako Pure Chemical) and oleoyl chloride (4.5 μL, 13.5 μmol; FUJIFILM Wako Pure Chemical) on ice, and incubated at room temperature for 48 hr while being mixed (*Figure 4—figure supplement 1*). The octadec-10-enoic-22-(octadic-10-enoyloxy)-docosanoic anhydride thus produced was hydrolyzed by incubating it with 200 μL of saturated aqueous sodium bicarbonate at room temperature for half a day, producing sodium 22-(octadec-10-enoyloxy)-docosanoate. Next, (*O*-C18:1)-ω-OH C22:0 FA was generated by adding 1 M hydrochloric acid to the above sodium salt until the pH of the sample was less than 3. Phase separation was performed by adding 300 μL of chloroform to the reaction solution, followed by centrifugation at 20,400 × g for 3 min at room temperature. The organic phase containing the OAHFA was recovered and dried.

The type 2ω WdiE hexadecane-1,16-diyl dioleate ([1,ω-*O*-C18:1]-C16:0) and type 2α WdiE 1-([hexadec-8-enoyl]oxy)-octadecan-2-yl oleate ([1,α-*O*-C18:1]-C16:0) were synthesized using the following procedures. 1,16-Hexadecanediol (12.2 mg, 47.2 μmol; Tokyo Chemical Industry) and 1,2-hexadecanediol (11.2 mg, 43.3 μmol; Tokyo Chemical Industry) were dissolved in 500 μL of 1-methyl-2-pyrrolidone, mixed with 1 mg (8.2 μmol) of 4-dimethylaminopyridine (FUJIFILM Wako Pure Chemical) and oleoyl chloride (100 μL, 298.9 μmol) on ice, and incubated overnight at room temperature while being mixed, generating (1,ω-*O*-C18:1)-C16:0 WdiE and (1,α-*O*-C18:1)-C16:0 WdiE, respectively (*Figure 5—figure supplement 1*). To dissolve the reaction products, 400 μL of ethyl acetate was added. Then, 400 μL of water was added for phase separation, followed by centrifugation (20,400 × g, 3 min, room temperature). The organic phases containing type 2α and type 2ω WdiEs were recovered and dried.

## Quantification and statistical analyses

Data are presented as means ± SD. The significance of differences between groups was evaluated using Student's *t*-test in Microsoft Excel (Microsoft, Redmond, WA) or Tukey-Kramer's test in JMP 13 software (SAS Institute, Cary, NC). A *P*-value of <0.05 was considered significant.

## Acknowledgements

We thank Dr Satoshi Ichikawa and Dr Akira Katsuyama (Hokkaido University) for their technical support. This work was supported by the Advanced Research and Development Programs for Medical Innovation (AMED-CREST), Grant Number JP19gm0910002h0005 (to AK) from the Japan Agency for Medical Research and Development (AMED), and by the Japan Society for the Promotion of Science (JSPS), KAKENHI, Grant Number JP18H03976 (to AK).

## Additional information

### Funding

| Funder | Grant reference number | Author |
|---|---|---|
| Japan Agency for Medical Research and Development | 19gm0910002h0005 | Akio Kihara |
| Japan Society for the Promotion of Science | JP18H03976 | Akio Kihara |

The funders had no role in study design, data collection and interpretation, or the decision to submit the work for publication.

### Author contributions

Masatoshi Miyamoto, Formal analysis, Investigation, Visualization, Writing - original draft; Takayuki Sassa, Resources; Megumi Sawai, Methodology; Akio Kihara, Conceptualization, Supervision, Funding acquisition, Writing - original draft, Project administration, Writing - review and editing

### Author ORCIDs

Masatoshi Miyamoto (iD) https://orcid.org/0000-0002-7785-7438
Takayuki Sassa (iD) https://orcid.org/0000-0003-3145-9829
Megumi Sawai (iD) http://orcid.org/0000-0002-7406-802X
Akio Kihara (iD) https://orcid.org/0000-0001-5889-0788

### Ethics

Animal experimentation: All animal experiments were approved by the institutional animal care and use committee of Hokkaido University (Permit Number: 17-0017).

### Decision letter and Author response

Decision letter https://doi.org/10.7554/eLife.53582.sa1
Author response https://doi.org/10.7554/eLife.53582.sa2

## Additional files

### Supplementary files

- Supplementary file 1. List of oligonucleotides used.

- Supplementary file 2. Selected *m/z* values for AMPP-OAHFAs in MS/MS analysis.

- Supplementary file 3. Selected *m/z* values for type 2ω/α WdiEs in MS/MS analysis.

- Supplementary file 4. Selected *m/z* values for type 1ω WdiEs in MS/MS analysis.

- Supplementary file 5. Selected *m/z* values for Chl-OAHFAs in MS/MS analysis.

- Supplementary file 6. Selected *m/z* values for CEs in MS/MS analysis.
- Transparent reporting form

## Data availability

All data generated or analysed during this study are included in the manuscript and supporting files.

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
