## [Decision Letter]

**Acceptance summary:**

Meibomian glands are located along the edge of eyelids and produce meibum, a lipid-rich secretion that protects us from dry eye disease by preventing evaporation of the eye's tear film. This paper describes a crucial role for a fatty acid omega hydroxylase, *Cyp4f39*, in producing O-acyl-omega-hydroxy fatty acids (OAHFAs), a class of lipids that ensures proper functioning of meibum. The study combines the development of elegant genetic models along with the synthesis of new lipid standards for quantifying OAHFA levels to deliver new insights into dry eye disease.

**Decision letter after peer review:**

Thank you for submitting your article "Lipid polarity gradient formed by ω-hydroxy lipids in tear film prevents dry eye disease" for consideration by *eLife*. Your article has been reviewed by three peer reviewers, and the evaluation has been overseen by a Reviewing Editor and Suzanne Pfeffer as the Senior Editor. The reviewers have opted to remain anonymous.

The reviewers have discussed the reviews with one another and the Reviewing Editor has drafted this decision to help you prepare a revised submission.

Summary:

Human dry eye diseases are major discomforting ocular conditions that affect millions of people worldwide. Lipids in the tear film layer provide a barrier against evaporation and thus prevent evaporative dry eye. In this study, the authors describe a role for *Cyp4f39*, a fatty acid omega hydroxylase, in the biosynthesis of O-acyl-omega-hydroxy fatty acids (OAHFAs), a class of lipids that is crucial for maintaining the proper polarity of meibum lipids in the tear film. Using genetically modified mice and several elegant technical advances, they show that *Cyp4f39* deficiency in the meibomian glands leads to OAHFA deficiency and human dry eye like pathology in mice. All reviewers expressed enthusiasm for this study and felt that it represented a significant advance in dry eye disease research. They did, however, raise some points that need to be addressed, both to support the conclusions of this study and to clarify the findings for the broader audience of *eLife*. Please prepare a revised version for resubmission where the following points are addressed.

Essential revisions:

1) Some of the major conclusions on mouse pathology of MGD need to be clarified:

a) Age: All the mouse pathological assessments are done at 12-14 months of age (Figure 1, 2), which is equivalent to mid-age human. Human dry eye disease has been shown to have an association with age. No description or discussion is found on whether this pathology in mice is age-dependent? When do the first symptoms appear?

b) Plugging of the meibomian gland orifices: no description is provided on how the obstruction is determined? Was it a qualitative or quantitative measurement? How many observations were made and at what age? Please provide better quality images for Figure 2D that more clearly show plugged vs normal meibomian glands.

c) Histological images of meibomian glands in Figure 3A need a clear description and annotation are needed on what is being shown; is it an acini? Please indicate by arrows the locations of the basal region, the central region, and enucleated cells. If possible, use a micrograph or cartoon of meibomian gland to explain these.

d) Results on corneal epithelial damage, in Figure 2C, which is higher in transgenic mice is contradictory to the Figure 3B, that shows normal 'formation' of cornea. If the authors wish to say the 'development' or formation is normal, they should show images corresponding to different age points, especially at early age points of 2-3 months when the development of mouse eye completes. In Figure 3B generated from 12-month-old mice, epithelial damage is expected.

2) A major connection appears to be missing regarding lipid composition and mouse pathology – if the pathology is obstruction of meibomian glands, then that would cause reduction in secretion of all lipids, not only OAHFA containing lipids. Is the mouse dry eye pathology observed here in the *Cyp4f39* knockout due to deficiency of all meibum lipids or only OAHFA containing lipids?

3) In Figures 1C, 2A, and 2E, the control group includes both *Tg-Cyp4f39^+/+^* and *Tg-Cyp4f39^+/-^* mice, whereas in the rest of the manuscript the control group is simply *Tg-Cyp4f39^+/+^*. Please either remove data for the *Tg-Cyp4f39^+/-^* mice from the control grouping in Figure 1C, 2A, and 2E, or separate into +/+, +/-, and -/- mice groupings. Theoretically, *Tg-Cyp4f39^+/-^* mice may not be completely "normal".

4) Subsection “Palpebral ptosis and abnormal tear covering on the eyeball surface in *Cyp4f39*-deficient mice”. Please make sure that 3 x FLAG-*Cyp4f39* protein is expressed in keratinized epithelial cells, but not in meibocytes, by immunohistochemistry.

5) Subsection “*Cyp4f39* deficiency causes a decrease in C16:1 OAHFA levels in meibum lipids”. Is it possible to separate keratinized epithelial cells and meibocytes by flow cytometry and confirm that C18:1 and C18:2 OAHFAs are indeed derived from keratinized epithelial cells?

6) It is interesting that OAHFAs in meibum lipids contain C16:1 in preference to C18:1. In most tissues and cells, C18:1 is more abundant than C16:1. Do meibocytes contain C16:1 abundantly? Or is there another explanation for the enrichment of C16:1 in OAHFAs? Given that PNPLA1, a transacylase, incorporates 18:2 selectively into w-hydroxy-VLCFA in ceramide to give rise to w-O-acylceramide, a similar reaction with C16:1 specificity could occur in OAHFA production in the meibomian glands? Please comment.

7) The statements "*Cyp4F39* is essential for WdiE production" or "OAHFAs produced by *Cyp4f39*" are somehow misleading, since the enzyme does not catalyze the esterification of OAHFAs. Strictly speaking, *Cyp4F39* produces OAHFA precursors (omega-hydroxy-VLCFAs), which are subsequently utilized for OAHFA synthesis. The enzyme involved in the esterification step is not described in this study. In this vein, please check the statements throughout the text so as to avoid confusion.

---

## [Author Response]

Essential revisions:1) Some of the major conclusions on mouse pathology of MGD need to be clarified:a) Age: All the mouse pathological assessments are done at 12-14 months of age (Figure 1, 2), which is equivalent to mid-age human. Human dry eye disease has been shown to have an association with age. No description or discussion is found on whether this pathology in mice is age-dependent? When do the first symptoms appear?

We examined the age-dependent number of blinks (Figure 2A). *Tg-Cyp4f39*^–/–^ mice tended to blink more frequently as they grew older, but there were no statistically significant changes with age. Based on this result, we concluded that there is no clear age-dependent progression of dry eye (subsection “*Cyp4f39* deficiency causes dry eye due to plugging of the meibomian gland orifices”). Regarding plugging of the meibomian gland orifice, only data from 14-month-old *Tg-Cyp4f39*^–/–^ mice were presented in the original manuscript. In the revised manuscript, we have added data for 6-month and 12-month-old mice. As shown in the new Figure 2D, there are no obvious differences in the degree of plugging between the age groups. This result supports our conclusion that there is no clear age-dependent progression of dry eye in *Tg-Cyp4f39*^–/–^ mice.

b) Plugging of the meibomian gland orifices: no description is provided on how the obstruction is determined? Was it a qualitative or quantitative measurement? How many observations were made and at what age? Please provide better quality images for Figure 2D that more clearly show plugged vs normal meibomian glands.

Plugging was judged by observing the meibomian glands prepared by dissection. In *Tg-Cyp4f39*^–/–^ mice, plugging was observed in 8 out of 8 mice aged 6 months or older (6 months, 2 mice; 12 months, 3 mice; 14 months, 1 mouse; and 17 months, 2 months). We did not examine mice younger than 6 months for evidence of plugging. Therefore, plugging occurs at least at 6 months of age. On the other hand, no plugging was observed in control mice between 6–12 months (0 out of 9 mice). These descriptions are included in the revised manuscript. An enlarged view has also been added to make the plugging more clearly visible (Figure 2D).

c) Histological images of meibomian glands in Figure 3A need a clear description and annotation are needed on what is being shown; is it an acini? Please indicate by arrows the locations of the basal region, the central region, and enucleated cells. If possible, use a micrograph or cartoon of meibomian gland to explain these.

A schematic diagram illustrating the cells of the meibomian gland acini has been added to Figure 3A.

d) Results on corneal epithelial damage, in Figure 2C, which is higher in transgenic mice is contradictory to the Figure 3B, that shows normal 'formation' of cornea. If the authors wish to say the 'development' or formation is normal, they should show images corresponding to different age points, especially at early age points of 2-3 months when the development of mouse eye completes. In Figure 3B generated from 12-month-old mice, epithelial damage is expected.

The corneal damage in Figure 2C was observed to have a scattered distribution across the eye, which corresponds to the pattern of damage seen in human dry eye. Figure 3B does not show the damaged part of cornea; rather, it shows a normal part to indicate that there was no abnormality in corneal development. Even in cases where morphological abnormalities were observed by HE staining, it was difficult to discriminate whether they represented true corneal damage or an artifact during the HE staining procedure.

2) A major connection appears to be missing regarding lipid composition and mouse pathology – if the pathology is obstruction of meibomian glands, then that would cause reduction in secretion of all lipids, not only OAHFA containing lipids. Is the mouse dry eye pathology observed here in the Cyp4f39 knockout due to deficiency of all meibum lipids or only OAHFA containing lipids?

We speculate that impaired production of OAHFA and OAHFA metabolites causes an abnormal extension of the lipid layer onto the aqueous layer, leading to abnormal secretion of meibum lipids from the orifice of the meibomian gland. We cannot exclude the possibility that the plugging in *Tg-Cyp4f39*^–/–^ mice reduces the secretion of other meibum lipids and causes further progression of dry eye. However, the plugging in *Tg-Cyp4f39*^–/–^ mice is mild (the plugs had a semi-liquid state), and the meibomian glands are not enlarged. If secretion of meibum lipids is strongly inhibited, the meibomian glands are expected to swell. Therefore, we speculate that most meibum lipids are secreted in *Tg-Cyp4f39*^–/–^ mice. In fact, we possess certain mice (other genetically modified mice) that show a firm, solid-state plugging and have swollen meibomian glands (our unpublished data). The above description, except for the unpublished data, is described in the revised manuscript Discussion (paragraph six).

3) In Figures 1C, 2A, and 2E, the control group includes both Tg-Cyp4f39^+/+^ and Tg-Cyp4f39^+/-^ mice, whereas in the rest of the manuscript the control group is simply Tg-Cyp4f39^+/+^. Please either remove data for the Tg-Cyp4f39^+/-^ mice from the control grouping in Figure 1C, 2A, and 2E, or separate into +/+, +/-, and -/- mice groupings. Theoretically, Tg-Cyp4f39^+/-^ mice may not be completely "normal".

In the revised manuscript, the results obtained from *Tg-Cyp4f39^+/+^* and *Tg-Cyp4f39*^+/–^ mice are separately represented. There were no differences between *Tg-Cyp4f39^+/+^* and *Tg-Cyp4f39*^+/–^ mice with regards to any of the results.

4) Subsection “Palpebral ptosis and abnormal tear covering on the eyeball surface in Cyp4f39-deficient mice”. Please make sure that 3 x FLAG-Cyp4f39 protein is expressed in keratinized epithelial cells, but not in meibocytes, by immunohistochemistry.

In response to this comment, we performed the suggested immunohistochemistry experiment. However, we could not obtain clear results due to non-specific background signal. Since the origin of the residual band in Figure 1D is not an important issue and does not affect the conclusions of our manuscript, we did not examine it further.

5) Subsection “Cyp4f39 deficiency causes a decrease in C16:1 OAHFA levels in meibum lipids”. Is it possible to separate keratinized epithelial cells and meibocytes by flow cytometry and confirm that C18:1 and C18:2 OAHFAs are indeed derived from keratinized epithelial cells?

At present, it is not possible to perform flow cytometry in this case because no cell surface markers specific to meibocytes are known.

6) It is interesting that OAHFAs in meibum lipids contain C16:1 in preference to C18:1. In most tissues and cells, C18:1 is more abundant than C16:1. Do meibocytes contain C16:1 abundantly? Or is there another explanation for the enrichment of C16:1 in OAHFAs? Given that PNPLA1, a transacylase, incorporates 18:2 selectively into w-hydroxy-VLCFA in ceramide to give rise to w-O-acylceramide, a similar reaction with C16:1 specificity could occur in OAHFA production in the meibomian glands? Please comment.

We examined the fatty acid composition of meibum lipids and found that C18:1 fatty acid was more abundant than C16:1 fatty acid (New Figure 4G), suggesting that the unknown mouse acyltransferase involved in OAHFA production prefers C16:1 acyl-CoA as a substrate rather than C18:1-CoA.

Acylceramides, which are important for the skin barrier, consist of a long-chain base, an ω-hydroxy fatty acid, and linoleic acid; the conjugate of the latter two components is C18:2 OAHFA. PNPLA1 catalyzes the formation of an ester bond between the ω-hydroxy fatty acid and linoleic acid. C18:2 OAHFA present in the epidermis is considered to be generated by the degradation of acylceramide. In the present study, we detected C18:2 OAHFA in meibum lipids. This may be derived from the keratinized epithelial cells constituting the meibomian gland ducts or from epidermis contamination in the samples. On the other hand, since PNPLA has low activity toward C16:1 fatty acid, it is unlikely that PNPLA is involved in C16:1 OAHFA production.

7) The statements "Cyp4F39 is essential for WdiE production" or "OAHFAs produced by Cyp4f39" are somehow misleading, since the enzyme does not catalyze the esterification of OAHFAs. Strictly speaking, Cyp4F39 produces OAHFA precursors (omega-hydroxy-VLCFAs), which are subsequently utilized for OAHFA synthesis. The enzyme involved in the esterification step is not described in this study. In this vein, please check the statements throughout the text so as to avoid confusion.

We believe that the statement "Cyp4f39 is involved in or essential for the production of OAHFA and OAHFA derivatives" is appropriate, since these lipids cannot be produced in the absence of Cyp4f39. The phrase “Cyp4F39 produces OAHFA precursors (ω-OH VLCFAs), which are subsequently utilized for OAHFA synthesis” is long. Therefore, we would like to avoid such phrasing to prioritize readability. Regarding the phrase “OAHFAs produced by Cyp4f39”, we agree that there is a possibility it could be misread. We have therefore changed it to “Since Cyp4f39 is involved in the production of OAHFAs composed of a C30–C36 ω-OH FA and a C16:1 FA (Figure 4C) […]”.